# Red LED Light Irradiation Increases the Resistance Against Environmental Stress of Frozen Bovine Sperm Thawed in Suboptimal Conditions

**DOI:** 10.3390/ani15223353

**Published:** 2025-11-20

**Authors:** Olga Blanco-Prieto, Carolina Maside, Andrea Gruzmacher, Manuel Ortiz, Marcelo Ratto, Francisco Javier Urra, Tomás Vera, Pablo Strobel, Jaime Catalán, Beatrice Mislei, Diego Bucci, Marc Yeste, Joan E. Rodríguez-Gil, Alfredo Ramírez-Reveco

**Affiliations:** 1Department of Animal Medicine and Surgery, Faculty of Veterinary Medicine, Autonomous University of Barcelona, ES-08193 Bellaterra, Spain; olga.blanco@udl.cat; 2Department of Animal Science, University of Lleida, ES-25198 Lleida, Spain; 3Biotechnology of Animal and Human Reproduction (TechnoSperm), Institute of Food and Agricultural Technology, University of Girona, ES-17003 Girona, Spainjaime.catalan@uab.cat (J.C.); marc.yester@udg.edu (M.Y.); 4Unit of Cell Biology, Department of Biology, Faculty of Sciences, University of Girona, ES-17003 Girona, Spain; 5Artificial Insemination Center (CIA), Center for Capacitation and Training in Animal Handling and Reproduction (CENEREMA), Valdivia CL-5091000, Chilemarceloratto@uach.cl (M.R.); pablostrobel@uach.cl (P.S.); alfredoramirez@uach.cl (A.R.-R.); 6Institute of Animal Science, Faculty of Veterinary Medicine, Austral University of Chile, Valdivia CL-5091000, Chile; manuelortiz@uach.cl; 7National Institute for Artificial Insemination (AUB), University of Bologna, IT-40057 Cadriano, Italy; beatrice.mislei@unibo.it; 8Department of Veterinary Medicine Science, University of Bologna, IT-40126 Ozzano dell’Emilia, Italy; diego.bucci@unibo.it; 9Catalan Institution for Research and Advanced Studies (ICREA), ES-08010 Barcelona, Spain

**Keywords:** red light, irradiation, cryopreserved sperm, artificial insemination

## Abstract

The thawing of commercial frozen bull straws through red LED irradiation in suboptimal conditions (environmental temperature: 20 °C; PHOTO) induced an overall slight impairment of in vitro semen quality when compared with samples thawed in standard conditions. However, the overall in vitro quality of PHOTO samples was better than that of those thawed in the same suboptimal conditions but without light irradiation (ET). This indicates that light irradiation could increase the resiliency of samples to thawing in non-optimal environmental conditions. Furthermore, preliminary, purely descriptive tests in in-farm conditions suggest that red light irradiation could increase in vivo pregnancy rates, especially when non-optimal thawing conditions are applied at the farm, although further studies involving more extensive in-farm trials are needed to completely confirm this assertion.

## 1. Introduction

Freezing and thawing involve a series of stressful events that can harm sperm cells. Previous research demonstrated that many compartments/components of the sperm cell can be damaged after cryopreservation, which may ultimately have a detrimental impact on its fertilizing ability [1,2]. The potential reduction in post-thaw sperm function and survival must be particularly considered in the case of artificial insemination (AI) with frozen–thawed sperm, as it is a very important tool to optimize reproductive performance when selecting breeds or conserving endangered species [3,4]. In cattle, AI with commercial frozen–thawed sperm is routinely used for breeding [3,5]. Calving rates are, however, low when utilizing frozen–thawed sperm [6]. This relatively low efficiency arises from different factors, including the often-negative energy balance of dairy cows, the insemination skills of practitioners, and the variations in the recommended thawing protocol (i.e., incubation at 38 °C for 40 s), which is often impossible to apply in farm conditions (see [7] as a review). Hence, any effort to increase AI outcomes in suboptimal conditions, such as those often yielded in in-farming situations, should deserve attention.

Irradiation of mammalian sperm with red light has been reported to improve sperm quality, resilience to thermal stress, and fertilizing ability linked to a direct impact on mitochondrial function. In effect, stimulation of pig sperm with red LED light improves their ability to withstand thermal stress at 37 °C for 90 min [8]. Likewise, red LED light irradiation increases the resilience of frozen–thawed horse sperm to post-thaw incubation at 38 °C for 120 min [9]. On the other hand, AI results obtained after red LED light irradiation of liquid preserved pig semen were found to be inconsistent, ranging from a significant increase in in vivo fertility and prolificacy [8,10] to no changes in these parameters [10,11]. Several factors could contribute to explaining these disparate outcomes, including farming and environmental conditions, the breed, and the specific irradiation protocol, among others [8,10,11]. Likewise, there is precise knowledge regarding the mechanisms by which red LED light can act on sperm function. In this respect, there are at least three possible action pathways. The first mechanism would involve a direct action of light on chemical bonds, such as phosphodiester ones of ATP molecules, on which irradiation would increase the energy level accumulated in these bonds [12]. A second mechanism would be linked to the presence of light-sensitive molecules in the sperm cell membrane. In this sense, opsins like rhodopsin, which is sensitive to red light, have been found in mammalian sperm membranes [13]. The third pathway would imply the light-induced activation of the mitochondrial electron chain, which has been observed in boars [14], horses [15], and donkeys [16]. Thus, the precise balance of all these mechanisms, together with others that are not known at this moment, could explain the effects of red LED light depending on the exact status of sperm function during irradiation. At present, there is no data on the effects of sperm irradiation upon AI outcomes in cattle. Yet, previous research in other species suggests that thawing bovine sperm straws with red LED light could have a positive effect on in vivo fertility in cattle, especially in suboptimal conditions. The successful application of irradiated semen for AI, nevertheless, needs to be optimized in every mammalian species.

This study sought to address, for the first time, whether thawing commercial, cryopreserved bovine sperm through irradiation with red LED light (in suboptimal conditions) could improve the resilience of cells to the induced thawing-related environmental stress. Furthermore, a small, preliminary, and non-conclusive test involving AI was performed to do a first exploration regarding the possibility that red LED irradiation could have an impact not only on post-thawing sperm quality and functionality, but also on in vivo reproductive performance after AI in farm conditions.

## 2. Materials and Methods

### 2.1. Animals and Samples

Frozen semen straws from 16 different bulls (4–8 years old) provided by the Artificial Insemination Center (CIA), Austral University of Chile, were used. The straws (concentration: 10–20 × 10^6^ sperm) were thawed through three different protocols. In protocol 1 (Irradiated Group; PHOTO), frozen straws were placed inside a specifically designed red LED light stimulation device (MaxiCow^®^; Barcelona, Spain) and irradiated for 5 min (2 min light, 1 min darkness, and 2 min light) at an environmental temperature of 20 °C. The light intensity was 5.66 mW/cm^2^. In protocol 2 (Environmental Temperature Group; ET), the straws were placed inside a MaxiCow^®^ device for 5 min at an environmental temperature of 20 °C, but with the irradiation system switched off. In protocol 3 (Control Group; CONTROL), the straws were thawed by soaking in a water bath at 38 °C for 40 s. The temperature variation was evaluated in 6 different straws from each of the thawing protocols.

Immediately after thawing, 4–5 thawed straws belonging to the same ejaculate to avoid masking of inter-bull variability were pooled and immediately used. The total number of utilized straws, summing up all experiments, was 260. For temperature records, empty semen straws were loaded, sealed, cryopreserved, and further thawed with the three described protocols, while being in contact with a probe coupled to a USB data logger (ThermoWorks, Alpine, UT, USA). The data were plotted and analyzed with GraphPad Prism Ver. 6 (GraphPad Software, San Diego, CA, USA). For protocols 1 (PHOTO) and 2 (ET), the temperature in and around the MaxiCow^®^ device was found to be around 20 °C.

Regarding sperm analysis, the pooled samples were split into three aliquots. The first aliquot was used to evaluate sperm motility, acrosome integrity, mitochondrial membrane potential, DNA integrity, and intracellular levels of reactive oxygen species (ROS) through flow cytometry. The second aliquot was intended for a chromatin condensation assay. For this purpose, the aliquot was centrifuged at 900× *g* for 30 s at 20 °C, and the resulting pellet was immediately frozen by immersion into liquid N_2_ and subsequently stored at −80 °C until use. The third aliquot was utilized to perform both nitro blue tetrazolium chloride (NBT) and resazurin tests. These samples were washed twice with a buffer solution (pH 7.4) of 300 mM Tris-HCl and 94.7 mM citrate. Then, the cells were centrifuged at 540× *g* and 20 °C for 10 min, and the resulting pellets were resuspended and adjusted to a concentration of 65 × 10^6^ sperm/mL.

### 2.2. Evaluation of Post-Thaw Sperm Integrity and Functionality

#### 2.2.1. Sperm Viability

Sperm viability was determined through the acridine orange (AO) and propidium iodide (PI) stains as described in [17] with minor modifications. Briefly, a 3 μL drop was mixed (1:1, *v*:*v*) with a staining solution of 10 μM AO and 30 μM PI in TBS buffer (pH 7.4) on a microscope slide at 37 °C. The samples were subsequently analyzed using a computer-assisted sperm analysis (CASA) system (Viability Module; Sperm Class Analyzer^®^, Microptic S.L., Barcelona, Spain) coupled to an epifluorescence microscope (Nikon E200; Tokio, Japan) with a high-velocity video recorder (Basler scA780 54tc; Basler AG, Ahrensburg, Germany). Five hundred sperm cells were counted per sample and automatically classified as viable (green sperm) or non-viable (red or orange sperm).

#### 2.2.2. Acrosome Integrity

Acrosome integrity was examined through Coomassie G-250 staining (Sigma-Aldrich; Saint Louis, MO, USA), following the protocol set by Larson and Miller (1999) [18]. Briefly, 5 μL of each sample was smeared and dried onto a slide. Slides were then fixed with 100% (*v*:*v*) methanol at 4 °C for 30 min, air-dried, and subsequently stained with a Coomassie blue solution (0.22% G-250, 50% methanol, and 10% glacial acetic). After incubation at room temperature for 15 min, the samples were thoroughly washed with double-distilled water, dried at 20 °C, and then evaluated under a bright-field microscope (Nikon E200) at 1000× magnification. The percentage of acrosome-intact sperm was calculated after performing two counts of 150 sperm each.

#### 2.2.3. Sperm Motility

Sperm motility was assessed with a CASA system (Motility Module; Sperm Class Analyzer^®^, Microptic S.L.), according to Ramírez-Reveco et al., 2016 [19] and Miguel-Jiménez et al., 2020 [20]. In brief, 3 μL of each sample was placed into a prewarmed (38 °C) Leja^®^ Standard count four-chamber slide 20 µm (Leja Products; Nieuw Vennep, The Netherlands). The samples were evaluated under a negative objective (10×) in a phase contrast microscope (Nikon E200, Tokyo, Japan), coupled to a high-velocity digitalized video recorder (Basler scA780 54tc; Basler AG). Motility evaluation was based on the analysis of 25 consecutive, digitalized photographic images taken over a time lapse of 1 s. A minimum of 200 sperm cells were examined. The following kinetic parameters were recorded: curvilinear velocity (VCL), linear velocity (VSL), average path velocity (VAP), linearity coefficient (LIN), straightness coefficient (STR), wobble coefficient (WOB), mean lateral head displacement (mALH), frequency of head displacement (BCF), dance (DNC), and mean dance (mDNC: LIN × mALH). A sperm cell was considered motile when VAP was at least ^3^ 10 µm/s, and progressively motile when STR was at least ^3^ 70%.

#### 2.2.4. Chromatin Condensation

Chromatin condensation was analyzed as the integrity of disulfide bonds between cysteines in nucleoproteins [21] following the protocol described in [22]. In brief, samples were centrifuged at 2000× *g* and 16 °C for 30 s. The supernatants were discarded, and the pellets were resuspended in a cysteine buffer (50 mM Tris-HCl, 150 mM NaCl, 1% (*v*:*v*) Nonidet P-40, 0.5% (*w*:*v*) sodium deoxycolate 0.5%, 1 mM Na_2_VO_4_, 10 µmol/mL leupeptin, 1 mM benzamidine, and 0.5 mM phenylmethyl sulfonil fluoride (PMSF); pH 7.4). The samples were homogenized through sonication at 4 °C (50% amplitude, 11 long-lasting pulses; Bandelin Sonoplus HD 2070; Bandelin Electronic GmbH and Co., Berlin, Germany) and then centrifuged at 850× *g* and 4 °C for 20 min. The pellets were resuspended in 1 mL of cysteine buffer. Thereafter, 20 µL of each resuspended sample was mixed with 1 mL of 0.2 mM 2,2′-dipyrydil disulfide in PBS. The samples were incubated at 37 °C for 60 min and then examined with a spectrophotometer (WPA Lightwave II; Biochrom Ltd., Cambridge, UK) at 343 nm. The data were normalized against the total protein content of each sample, determined through a commercial kit (BioRad; Hercules, CA, USA) based on the Bradford method [23].

#### 2.2.5. Sperm DNA Fragmentation

Sperm DNA fragmentation was assessed with a modified alkaline Comet assay as described in [24] for pig sperm. Briefly, samples were adjusted to a final sperm concentration of 1 × 10^6^ sperm/mL in PBS. Afterwards, samples were mixed with 1% (*w*:*v*) low-melting-point agarose previously melted at 37 °C. Subsequently, 6.5 µL aliquots of the mixture were transferred onto agarose pre-treated slides and then covered with round coverslips. The samples were subsequently jellified by incubation at 4 °C for 5 min on a cold plate. Next, the slides were incubated at room temperature with three consecutive lysis solutions. First, samples were incubated with lysis solution 1 (0.8 M Tris-HCl, 0.8 M DTT, and 1% SDS; pH 7.5) for 30 min. Second, they were incubated with lysis solution 2 (4 M Tris-HCl, 0.4 M DTT, 50 mM EDTA, 2 M NaCl, and 1% Tween20; pH 7.5) for 30 min. Finally, samples were incubated with lysis solution 3 (0.4 M Tris-HCl, 0.4 M DTT, 50 mM EDTA, 2 M NaCl, 1% Tween20, and 100 μg/mL Proteinase K; pH 7.5) for 180 min. The slides were subsequently washed in deionized water for 2 min and incubated at 20 °C in a cold alkaline solution (0.03 M NaOH, 1 M NaCl) for 2.5 min and then with 0.4 M Tris-HCl (pH 7.5) for 4 min. The slides were subjected to electrophoresis (21 V; 1 V/cm) for 4 min through an alkaline electrophoresis buffer (0.03 M NaOH; pH 13). After electrophoresis, slides were dehydrated through an ethanol series (70%, 90% and 100%; 2 min each step) and then dried.

Finally, the slides were observed under a Zeiss Imager Z1 epifluorescence microscope (Carl Zeiss AG, Oberkochen, Germany), and 100 sperm per sample were examined. Different pictures were taken and analyzed with a CometScore v2.0 (RexHoover). The analysis of Comet heads and tails was performed automatically after adjusting the background intensity and correcting for detection errors and overlapping. In this technique, fragmented DNA is displayed as a stained area around the head that is displaced in the direction of the electrophoresis. This displacement formed a tailed area that vaguely resembles an olive form. In this manner, the area showing fragmented DNA was named the olive tail.

#### 2.2.6. Antioxidant Capacity

Antioxidant capacity of sperm was evaluated with the resazurin test, as described in [25] with modifications. Briefly, the pellets were resuspended in 495 µL of a Tris-citrate buffer (330 mM Tris, 94.7 mM citric acid; pH: 7.4) and added with 5 µL of 1.8 mM resazurin in PBS (pH 7.4). The samples were incubated at 37 °C for 45 min, prior to adding 500 mL of n-butyl alcohol. The specimens were then vortexed and centrifuged at 10,000× *g* for 5 min. The supernatants were recovered and used to evaluate absorbance at 610 nm (resazurin) and 570 nm (resofurin) with a spectrophotometer (UVmini-1240, Shimadzu^TM^; Kyoto, Japan). The results are given as A_570_/A_610_.

#### 2.2.7. Nitro Blue Tetrazolium Chloride Test

Total levels of reactive oxygen species (ROS) in sperm were determined through the evaluation of nitro blue tetrazolium chloride (NBT), which is oxidized by free oxidant radicals, following the protocol of [26] with minor modifications. Briefly, the pellets were resuspended in 500 µL of 1 mg/mL NBT in PBS and then incubated at 37 °C for 45 min. The samples were washed twice with Tris-citrate (330 mM Tris; 94.7 mM citric acid; pH 7.4) and further centrifuged at 540× *g* and 20 °C for 10 min. Following this, the pellets were resuspended in 120 µL of 2 M KOH in dimethyl sulfoxide. The absorbance of reduced NBT (formazan) was measured with a spectrophotometer at 630 nm (HR801, Shenzhen Highcreation Technology Co., Ltd., Shenzhen, China).

#### 2.2.8. Mitochondrial Membrane Potential

Mitochondrial membrane potential (MMP) was determined through JC-1 and SYBR-14 staining. In brief, the samples were washed through centrifugation at 800× *g* and room temperature for 2 min, and resuspended in Tyrode’s medium. Following this, sperm were incubated with 8.3 mM JC-1, 0.56 mM SYBR-14, and 16.7 mM propidium iodide at 38 °C for 20 min in the dark. Ten µL of each sperm suspension was placed onto a slide, and at least 200 sperm cells per sample were examined under a Nikon Eclipse E600 epifluorescence microscope (Nikon Europe BV, Badhoeverdop, The Netherlands). Sperm were classified into one of the four subpopulations: (i) viable sperm with active mitochondria (sperm head stained in green and mitochondria stained in orange); (ii) viable sperm with inactive mitochondria (both sperm head and mitochondria stained in green); (iii) non-viable sperm with active mitochondria (sperm head stained in red and mitochondria stained in orange); and (iv) non-viable sperm with inactive mitochondria (sperm head stained in red and mitochondria stained in green).

### 2.3. Flow Cytometry Analyses

Flow cytometry analyses were run to evaluate mitochondrial activity and cellular ROS production. All reagents were purchased from Thermo Fisher Scientific (Waltham, MA, USA). The analyses were performed following the indications of the International Society for Advancement of Cytometry [27]. In each assay, sperm concentration was set to 1 × 10^6^ sperm/mL in a final volume of 0.5 mL Tyrode’s medium. Once stained, samples were passed through a FACSCalibur flow cytometer (Becton Dickinson, Milan, Italy) equipped with a 488 nm argon-ion laser and a 635 nm red diode laser. The emission of each fluorochrome was detected with four filters: 530/30 band pass (green/FL1), 585/42 band pass (orange/FL2), >670 long pass (far red/FL3), and 661/16 band pass (orange for red laser/FL4). The data were acquired using the BD Cell Quest Pro software 5.1 (Becton Dickinson). Fluorescent signals were logarithmically amplified, and photomultiplier settings were adjusted to each specific staining method. The side scatter height (SSC-H) and forward scatter height (FSC-H) were recorded in linear mode (FSC vs. SSC dot plots), and the sperm population was positively gated based on morphologic characteristics. A minimum of 10,000 gated sperm events were evaluated per replicate. In CM-H_2_DCFDA, DHE, and MitoSOX assessments, percentages of non-DNA-containing particles (debris) were determined to avoid an overestimation of sperm particles as described in [28]. Examples of representative plots for flux cytometer analysis are shown in Appendix A.

#### 2.3.1. Evaluation of Intracellular ROS Levels (DHE/YO-PRO-1 and CM-H_2_DCFDA/PI)

Dihydroethidium (DHE) is a cell-permeable dye emitting in blue when not oxidized and in orange/red (610 nm) when oxidized mainly by superoxide radicals (O_2_^•−^). In this way, DHE can be used as a direct marker for ROS levels and as an indirect marker for the superoxide ones. YO-PRO-1 was used as an indicator of sperm with early apoptotic-like changes in membrane permeability.

CM-H_2_DCFDA is a non-fluorescent dye that accumulates in the cell cytoplasm due to deacetylation and emits green fluorescence after oxidation by an ROS and subsequent conversion into 2′,7′-dichlorofluorescein (DCF). This fluorochrome was coupled with PI, which stains non-viable sperm, emitting red/orange fluorescence. In brief, sperm samples were diluted in 500 mL of Tyrode’s medium and stained with 2.5 µL of 50 mM CM-H_2_DCFDA in DMSO and 2.5 µL of 2.4 mM PI in water. The samples were incubated at 37 °C for 30 min in the dark. Finally, ROS production was assessed as DCF-fluorescence in viable sperm.

#### 2.3.2. Mitochondria-Associated ROS Production (MitoSOX/Mitotracker Deep Red/YO-PRO-1)

MitoSOX Red (MX) is a lipid-soluble, cell-permeable cation that selectively targets the mitochondrial matrix and can thus detect superoxide radicals (O_2_^•−^) specifically generated by mitochondria. The MX emits red fluorescence upon oxidation by O_2_^•−^. In addition, MitoTracker deep red (MT-DR) was included to simultaneously assess mitochondrial integrity. Finally, YO-PRO-1 was used as an indicator of sperm with early apoptotic-like changes in membrane permeability. Regarding the protocol, sperm samples were incubated with 1 µL of 1 mM MX in DMSO, 2.5 µL of 100 nM MT-DR in DMSO, and 2.5 µL of 100 nM YO-PRO-1 in DMSO at 37 °C for 30 min in the dark. The mitochondrial production of ROS by viable sperm with intact mitochondria was recorded.

### 2.4. Artificial Insemination Trials

Artificial insemination (AI) descriptive trials were conducted at two separate cattle farms (Farms 1 and 2). Geographical and technical farm characteristics such as geographical coordinates, altitude above the sea, temperature and humidity during AI, number and general breed characteristics of cows were recorded. In all cases, AI procedures were based on fixed-time insemination, through synchronization of estrus in cyclic animals by using a standard protocol routinely performed on each farm. Since the objective of this preliminary trial was merely descriptive, animals that showed evident signs of estrus were randomly split into two groups so that both the control group and the treatment group contained animals of all body conditions, number of births, and production level. One group was inseminated with 0.25 mL commercial straws previously thawed under standard conditions at 38 °C for 40 s (CONTROL group). The other group was inseminated with commercial straws thawed through irradiation for 5 min (2 min light, 1 min darkness, 2 min light; MaxiCow^®^) at the corresponding environmental temperatures found at each farm. No straws placed inside the MaxiCow^®^ device while switched off (hence lacking the photoirradiation application) were used. Otherwise, the same batch of doses from the same ejaculate was utilized for replicates for AI. The cows were inseminated with standard intrauterine insemination, following the routine procedure utilized at each farm. Afterwards, pregnancy rates were recorded using standard transrectal ultrasonography after 26 days of AI; this was repeated 32 days after AI. Based upon the preliminary, descriptive nature of these trials, the farms were evaluated individually based on their type of production, thus combining the two control and treatment groups for dairy or beef cows separately. In sum, the total number of inseminated animals was 294. No further statistical analyses were performed, taking into account the very preliminary, purely descriptive nature of the test.

### 2.5. Statistical Analyses

Statistical analyses were conducted using a statistical package (SPSS^®^ Ver. 25.0 for Windows; IBM corp., Armonk, NY, USA). The data were first tested for normality and homoscedasticity through the Shapiro–Wilk and Levene tests, respectively. When required, the data were transformed through either log or arcsine √x.

The effects of thawing sperm through irradiation on sperm viability, acrosome integrity, sperm motility, chromatin condensation, DNA fragmentation, NBT, resazurin, MMP, and ROS levels were evaluated with one-way analysis of variance (ANOVA) followed by post hoc Sidak’s test for pair-wise comparisons. The warming rates (slopes) for treatments (control, thawing through light stimulation, thawing at 20 °C) were estimated by linear regression analysis of six replicates for each thawing treatment.

Motile sperm subpopulations were established as described in [29] with slight modifications. Briefly, all individual motion parameters taken from each individual sperm cell were utilized as independent variables in a Principal Component Analysis (PCA). The obtained matrix was rotated with the Varimax method and Kaiser normalization. The resulting regression scores were used to run a two-step cluster analysis based on the log-likelihood distance and Schwarz’s Bayesian Criterion. This analysis was performed separately for each experimental group (Control, ET, and PHOTO). In this way, cluster analysis for each group led every sperm cell to be classified in one specific subpopulation. The percentages of sperm belonging to each subpopulation in each experimental group were calculated per sample and compared through a one-way ANOVA followed by a post hoc Sidak’s test for pair-wise comparisons.

In all analyses, the level of significance was set at *p* < 0.05. Unless otherwise indicated, data are shown as mean ± standard error of the mean (SEM).

## 3. Results

### 3.1. Thawing Dynamics of Sperm Straws

CONTROL straws showed a biphasic curve, with a first, very fast thawing step from −196 °C to 35 °C at a rate of 1367.4 °C/min, followed by a second one from 35 °C to 38 °C at a rate of 0.8 °C/min (black slopes, Figure 1). On the contrary, PHOTO straws showed a less obvious biphasic curve, with a first thawing step from −196 °C to −10 °C at a rate of 270.4 °C/min, followed by a second step from −11 °C to 30 °C at a rate of 9.3 °C/min (red slopes, Figure 1). This led to reaching 0 °C about 60 s after the launch of the thawing protocol. Otherwise, the ET samples showed an even slower thawing curve, with a first step from −196 °C to −20 °C at a rate of 168.1 °C/min, and a second step from −20 °C to 20 °C at a rate of 10.0 °C/min (blue slopes, Figure 1). This led to reaching the temperature of 0 °C after about 120 s after the launch of the thawing protocol.

### 3.2. Effects of Red-Light Irradiation on Sperm Viability and Acrosome Integrity

Sperm viability in the PHOTO straws (mean ± SEM: 41.9% ± 4.1%) was lower (*p* < 0.05) than in both CONTROL (56.0% ± 4.3%) and ET ones (53.1% ± 2.2%; see Figure 2A). Otherwise, percentages of sperm with an altered acrosome did not differ between CONTROL and PHOTO straws (Figure 2B) but were significantly (*p* < 0.05) lower in ET samples (i.e., 7.9% ± 1.1% in ET vs. 10.6% ± 0.7% in PHOTO).

### 3.3. Effects of Red-Light Irradiation on Sperm Motility

Total motility was significantly higher (*p* < 0.05) in CONTROL samples than in both PHOTO and ET ones (CONTROL: 62.9% ± 5.2%, PHOTO: 44.3% ± 4.7%, ET temperature: 49.9% ± 3.2%; Figure 3). Otherwise, PHOTO samples yielded significantly (*p* < 0.05) greater values than those observed in ET ones (PHOTO: 23.5% ± 3.1%; ET: 19.4% ± 2.3%; Figure 3). When observing the mean motility parameters, the CONTROL group showed an overall better mean motion characteristics than those of both PHOTO and ET, although PHOTO samples yielded significantly (*p* < 0.05) higher values of mALH than CONTROL ones (PHOTO: 4.45 µm ± 0.02 µm; CONTROL: 4.16 µm ± 0.02 µm; Table 1). Otherwise, PHOTO samples showed significantly (*p* < 0.05) higher values of VCL, VSL, VAP, mALH, DNC, and mDNC than those of ET ones, whereas results of LIN and WOB were the contrary (Table 1).

### 3.4. Effects of Red-Light Irradiation on Motile Sperm Subpopulations

Four separate subpopulations were observed in all samples; these subpopulations were labelled from 1 (SP1) to 4 (SP4) following an ascending order based on their VAP (Table 2). The greatest subpopulation was SP1, whose proportions were 58.4% ± 9.5% in CONTROL, 58.3% ± 9.5% in PHOTO, and 69.7% ± 9.8% in ET samples; and the smallest one was SP4, whose proportions were 1.8% ± 0.4% in CONTROL, 1.5% ± 0.4% in PHOTO, and 0.2% ± 0.1% in ET (Figure 4). While no significant differences between CONTROL samples and PHOTO ones were observed in any of these four subpopulations, the proportions of sperm belonging to SP3 and SP4 were significantly (*p* ˂ 0.05) lower in ET straws than in those from both PHOTO and CONTROL (Figure 4). Despite this, kinetic parameters in each subpopulation differed between treatments. For example, all motion parameters of SP1, except VCL, mALH, and DNC, were significantly (*p* ˂ 0.05) lower in ET samples than in both CONTROL and PHOTO (Table 2). In addition, the BCF of SP1 was also higher (*p* ˂ 0.05) in CONTROL samples than in PHOTO ones. Regarding other subpopulations, when PHOTO samples were compared with ET ones, irradiated straws showed significantly (*p* ˂ 0.05) improved values of VCL, VSL, VAP, STR, mALH, and DNC of SP2; STR and mDNC of SP3, and all calculated parameters of SP4 (Table 2). When compared with CONTROL samples, PHOTO ones yielded worse significant (*p* ˂ 0.05) parameters of all parameters except mALH and DNC in SP2 and STR and DMC in SP3 (Table 2). The values of all parameters of SP4 were not significantly different between CONTROL straws and PHOTO ones (Table 2).

### 3.5. Effects of Red-Light Irradiation on Chromatin Condensation and DNA Fragmentation

Light irradiation during thawing in suboptimal conditions induced a significant (*p* < 0.05) improvement in values of chromatin decondensation when compared with ET straws (ET: 53.0 ± 4.3 µmol/mg protein vs. PHOTO: 36.5 ± 2.3 µmol/mg protein; Figure 5A), although values of PHOTO samples were in turn significantly higher than those of CONTROL ones (Figure 5A). It is worth noting that these findings concurred with the evaluation of sperm DNA fragmentation, as the mean olive tail moment was significantly (*p* ˂ 0.05) higher in ET sperm than in PHOTO samples (ET: 34.7 ± 1.3 arbitrary units vs. PHOTO: 16.1 ± 0.4 arbitrary units; Figure 5B). In fact, the results of PHOTO samples were even better than those of CONTROL ones, which yielded values of 11.0 ± 0.3 arbitrary units (Figure 5B).

### 3.6. Effects of Red-Light Irradiation on ROS Levels and Overall Antioxidant Capacity

When the antioxidant capacity evaluated as A_570_/A_610_ (resazurin test) was analyzed, no significant differences between the groups were detected (Figure 6A).

Total ROS levels evaluated with NBT were lower in both CONTROL and ET straws than in irradiated sperm (PHOTO: 0.698 ± 0.051 vs. CONTROL: 0.598 ± 0.050 and ET: 0.505 ± 0.012; Figure 6B). With regard to flow cytometry evaluations, the percentage of viable sperm with high DHE-marked ROS levels was low in CONTROL samples (1.1% ± 0.5%; Figure 7A) and did not differ from that found in ET ones. In contrast, thawing sperm with red light irradiation led to a small, although significant (*p* < 0.05), increase in the percentage of viable sperm with high ROS (mainly superoxide) levels (4.8% ± 1.5%; Figure 7A). Moreover, the percentage of viable sperm with high levels of total ROS determined by the H_2_DCFDA/PI test was very low without detecting any significant differences among groups (Figure 7B).

### 3.7. Effects of Red-Light Irradiation on Mitochondrial Function

Thawing cryopreserved sperm through both PHOTO and ET conditions was found not to affect mitochondrial membrane potential because the percentage of viable sperm with high mitochondrial membrane potential did not differ (*p* > 0.05) from CONTROL samples (Figure 8A).

Concerning flow cytometry analyses, the percentage of viable sperm with active mitochondria (MT-DR^+^/YO-PRO-1^−^) was 21.1% ± 6.1% in CONTROL samples (Figure 8B). This percentage did not significantly change in any of the other experimental conditions (Figure 8B). Moreover, neither the percentage of viable sperm with active mitochondria and high levels of superoxides originated from the mitochondria, nor the relative levels of mitochondria-formed superoxides were affected by thawing sperm in suboptimal conditions, either PHOTO or ET (Figure 9).

### 3.8. Effects of Red-Light Irradiation on In Vivo Reproductive Performance

Trials showed higher pregnancy values per AI (P-AI) in the group of cows inseminated with irradiated semen than in the CONTROL group at both farms. The maximum difference was observed in Farm 1 (CONTROL: 45.7%; P-AI, irradiated samples: 60.0%; see Table 3).

## 4. Discussion

Results shown in this study suggest that thawing cryopreserved bovine sperm in suboptimal conditions with the simultaneous application of a red LED light irradiation protocol has positive effects on sperm’s ability to withstand the rough environmental stress linked to the suboptimal thawing procedure. In this respect, it is known that slow thawing rates similar to those observed in both ET and PHOTO samples have been linked to a higher incidence of sperm DNA fragmentation [30], membrane damage [31], and greater ROS production and mitochondrial dysfunction [32]. All these alterations could be a consequence of an excessive formation of ice crystals and an increased osmotic stress associated with low thawing rates [33]. In addition to these facts, it was demonstrated that red light irradiation of pig semen increases its resilience to preservation at 16–17 °C in a commercial extender [34]. Similar effects have also been observed in other mammalian species [35]. This effect could be linked to a direct impact of red light on mitochondrial function, via photosensitizers of the electron chain [15,16,36]. Yet, mitochondrial function, which was evaluated through different techniques in the present study (mitochondrial membrane potential with JC1, mitochondrial integrity with MitoTracker^®^ Deep Red, and mitochondrial superoxide production with MitoSOX), was not found to be affected by light irradiation or thawing at room temperature. A similar lack of effect on mitochondrial function was also described in frozen horse sperm during freezing–thawing [9]. This could be because the thawing process by itself induces detrimental effects on sperm, so that changes in thawing conditions (i.e., conventional protocol, thawing through light irradiation, and thawing at room temperature) would not further interfere with mitochondrial function [37,38]. In addition, previous research showed that freeze–thawing of bovine sperm uncouples the mitochondrial bioenergetic profile [39], which differs from fresh semen, in which mitochondria are coupled [40]. Furthermore, Blanco-Prieto et al. [41] showed that total ROS levels determined through the DHE technique and mitochondria-originated ROS in frozen–thawed bovine sperm under standard conditions tend to increase in response to further post-thawing incubation rather than soon after thawing because of an impaired mitochondrial function (induced by specific complex inhibitors). This temperature-related effect could be similar to that observed in ET samples, which exhibited a different thawing temperature slope compared to the other experimental groups. Taking this into account, the existence of other, non-mitochondrial pathways by which red LED light can exert its effect gains importance. Thus, as mentioned in the Introduction, there are at least two other mechanisms by which red LED light can exert its effects on sperm. The best known is an opsin-related action, since opsins like rhodopsin have been found in mammalian sperm [13]. In this respect, the inhibition of the PKC-modulated opsin signalling pathway in freshly collected boar sperm has a minor effect on the response to red LED light when compared with the involvement of the mitochondrial electron chain [14]. However, this does not imply the absolute lack of an opsin-related light effect, and, in this way, opsin pathways could reach prominence when sperm mitochondrial activity was altered, as in freezing–thawing conditions. The other known mechanism that could increase its importance in explaining light effects during freezing–thawing was a direct action of red light on chemical bonds, such as the phosphodiester ones present in molecules like ATP, as previously described; inducing this mechanism increases the ability of phosphodiester bonds to accumulate energy [12]. In this way, the sum of mechanisms that are minoritarian in freshly obtained semen samples would achieve paramount importance when mitochondrial activity was altered by phenomena like freezing–thawing.

Analysis of sperm motility brought interesting results. While PHOTO samples showed significantly lower percentages of total and progressive motility and decreased kinetic parameters compared to CONTROL ones, the extent of that reduction was present even in ET straws. This could indicate that thawing samples through red light stimulation would induce a uniform, inhibitory effect on the motility of all sperm, notwithstanding that irradiation partially overcame the deleterious impact of thawing at a final temperature of 20 °C. Interestingly, thawing through irradiation was found to only affect the motion characteristics of two of the four identified motile sperm subpopulations, but not the percentages of sperm of each subpopulation. This result contrasts with that observed in ET samples, where not only the motion characteristics of each subpopulation but also the proportions of sperm included in subpopulations SP3 and SP4 were affected. This would suggest that neither all sperm respond to red LED light in the same manner, nor does thawing to a final temperature of 20 °C have the same effect on all cells. It is worth mentioning that irradiation of sperm with red light was also observed to affect motile subpopulations in other species, such as pigs [36] and horses [16], although in these studies, the utilized statistical approach did not allow for determining differences in the characteristics of each subpopulation. Herein, changes in sperm populations affected both the proportions and the specific motion profile of each subpopulation. Related to this and because, as aforementioned, red LED light was suggested to exert its effect mainly through direct stimulation of the mitochondria electron chain photosensitizers [16,36], it could be that the four subpopulations described would not only differ based on their motility but also on their mitochondrial sensitivity to light. In support of this prospect, sperm motility and mitochondrial activity were previously reported to be correlated in several mammalian species [42]. In this context, one should bear in mind that investigating sperm motility on the basis of subpopulations rather than looking into the entire population renders more accurate information regarding the functional status of the cell, as it considers the heterogeneous behaviour of sperm in a given ejaculate [43]. This individual approach is also relevant for the analysis of the sperm response to red light when used for thawing cryopreserved sperm.

The in vivo results shown here might be considered as purely preliminary, descriptive trials, which were devised to perform an initial, non-statistical approach to test if thawing sperm through light irradiation in farm conditions could affect semen fertilizing ability. In this way, no definite conclusion, by any means, can be yielded from the obtained results. This was due to the fact that the final number of animals that were available for us in this study was inevitably low, precluding an in-depth analysis of productive data. In addition, there are other reasons that preclude the presented in vivo study from being minimally conclusive. Thus, differences between farms in aspects such as environmental temperature, humidity, altitude above sea level, atmospheric conditions [44,45], farm management, and specific conditions of the inseminated cows, including breed, energy balance of the animal, age and number of parturitions [46], will affect the final outcomes of photostimulation. Moreover, one should take into consideration that while semen is known to play a key role in reproductive performance, it is not the sole factor, as the predictive value of the quality assessment of frozen–thawed bovine sperm with regard to the reproductive success has been found to be up to, but not greater than, 60% [47,48]. Then, as indicated above, results can be only taken as preliminary and, thus, their discussion is merely a first approach to the question. Taking this into account, it must be stressed that only two experimental groups were utilized (conventional thawing of straws, CONTROL vs. thawing through light irradiation). Otherwise, it is noteworthy that sperm thawed at environmental temperature without light irradiation were not included because it was not considered necessary for the main aim of the present work, which was to test the use of light irradiation under farm conditions. However, despite the inherently non-conclusive results of the in vivo trials, the results could suggest an apparent contradiction between the effects of light irradiation on in vitro sperm function parameters and in vivo sperm fertilizing ability. Always emphasizing the purely speculative basis, a logical explanation for these results would be that, after analyzing the absolute values of the analyzed in vitro parameters, the real number of sperm that are affected by freezing–thawing, following the results of in vitro tests, was not very high. In this way, whereas the absolute number of sperm that would colonize the oviduct in in vivo conditions was low (about 17,000; see [49]), both the CONTROL group and PHOTO samples have thousands to millions of sperm with acceptable functional characteristics, regardless of relative differences in the in vitro quality tests. Under these conditions, if all other factors modulating the success of AI are optimally conducted (i.e., properly conducted fixed-time AI protocols, optimal AI application, etc.), the impact of differences observed in in vitro analysis should not be significant. Otherwise, this can lead to another question, namely, why do in vivo results suggest an improving effect of red light irradiation, considering that the actual number of acceptable function sperm colonizing the oviduct was similar when comparing the CONTROL group and PHOTO ones? In this case, the explanation would be linked not to the absolute number of sperm arriving at the oviduct but to the functional status of these cells. Thus, in species such as boar, light exposition was able to increase the number of sperm that achieved the full capacitated status and, hence, increased the absolute number of sperm able to fertilize [50]. In fact, this effect was related to concomitant increases in in vivo fertility and prolificacy, although these results were variable, depending on factors such as specific farm management and environmental conditions [10,11]. In this way, considering the possibility that a similar increase in sperm ability to reach capacitation is induced in bull sperm—although the total number of sperm that colonize the oviducts was similar when comparing both the CONTROL and PHOTO samples—the greater ability to reach capacitation status would lead to a greater fertilizing ability efficiency after red LED light irradiation.

## 5. Conclusions

The present study suggests that thawing cryopreserved bovine sperm with red LED light in suboptimal conditions (final thawing temperature of 20 °C) significantly improved the overall in vitro semen quality when compared to straws thawed in the same conditions but without light irradiation. This indicates that red LED light exposition would improve the resistance of frozen bull semen when subjected to aggressive environmental conditions during thawing. Moreover, the preliminary in-farm AI results would open the possibility of optimizing sperm thawing protocols before AI in cattle by using light irradiation, especially if AI is performed in suboptimal conditions. Notwithstanding, taking into account the very preliminary results shown here, many additional studies are required to confirm the results obtained in vivo. These studies should be based on using a greater number of animals and farms in different management and environmental conditions before fully recognizing red LED light irradiation as a useful tool to improve AI fertility results in cow farms.

## 6. Patents

Method and apparatus improving the quality of mammalian sperm’ (European Patent Office No. 16199093.2; EP-3-323-289-A1.

## Figures and Tables

**Figure 1 animals-15-03353-f001:**
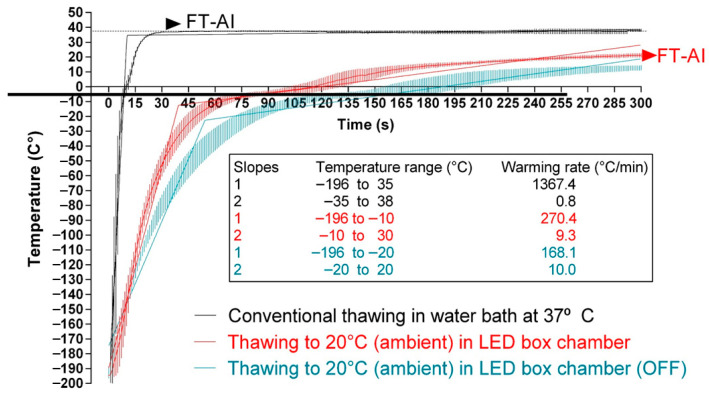
Thawing rates of cryopreserved bovine sperm subjected to standard thawing, irradiation with red LED light, or non-standard thawing at room temperature. The temperature was registered inside 0.25 mL straws with a probe. Black lines: straws thawed following the conventional protocol (38 °C for 40 s in a water bath; CONTROL). Red lines: straws thawed through irradiation with red light at room temperature (20 °C). Blue lines: straws thawed at room temperature (20 °C), inside the irradiation chamber but without any exposure to light. FT-AT: final temperature of the straw at the time of performing artificial insemination.

**Figure 2 animals-15-03353-f002:**
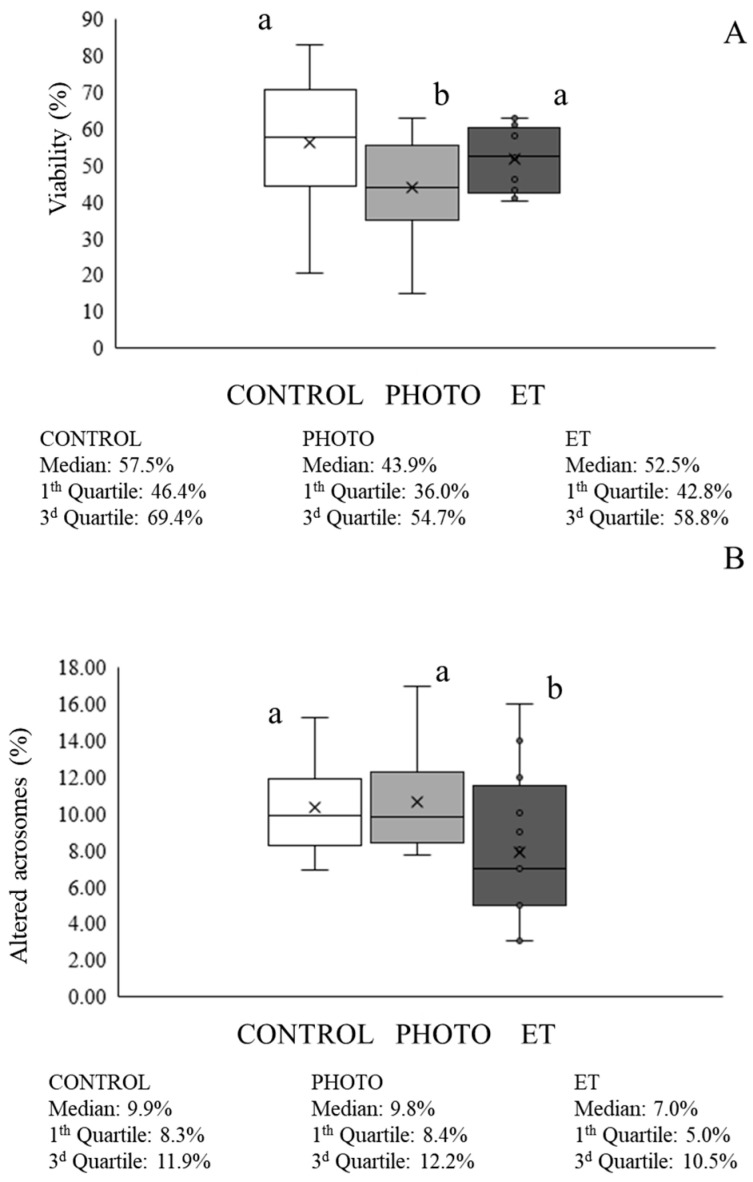
Sperm viability and acrosome integrity of cryopreserved sperm thawed through the conventional protocol, irradiation with red light at 20 °C, or thawed at 20 °C without irradiation. (**A**) Percentages of viable sperm. (**B**) Percentages of sperm with an altered acrosome. White bars (CONTROL): straws thawed following the conventional protocol (38 °C for 40 s in a water bath). Medium grey bars (PHOTO): straws thawed at 20 °C through irradiation with red light. Dark grey bars (ET): straws thawed at 20 °C inside the irradiation chamber but without any exposure to light. Results are expressed as box and whiskers; thus, median and quartiles (1st and 3rd) rather than means ± SEM are shown for each experimental point (Control, PHOTO, and ET) (n = 16). Different letters indicate significant (*p* < 0.05) differences between groups.

**Figure 3 animals-15-03353-f003:**
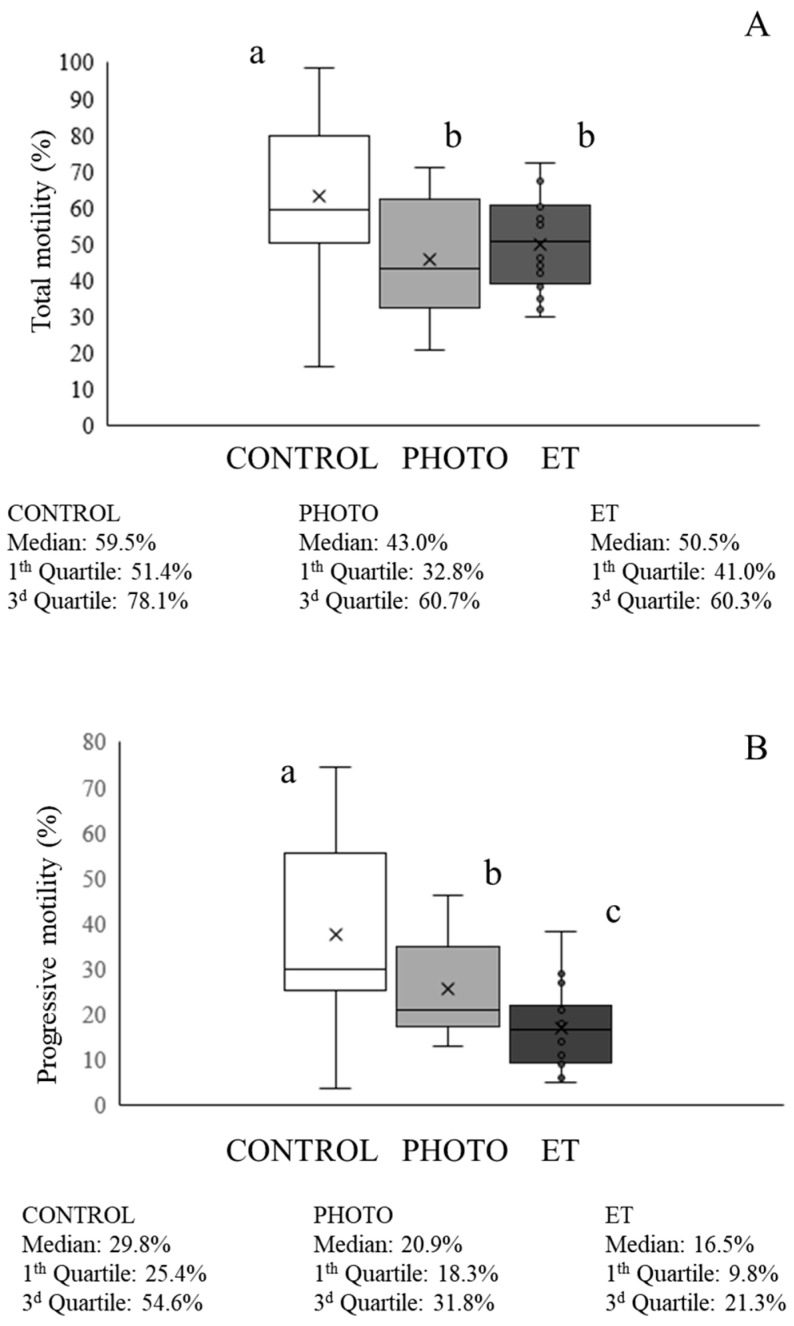
Total and progressive motility of cryopreserved sperm thawed through the conventional protocol, irradiation with red light at 20 °C, or thawed at 20 °C without irradiation. (**A**) Percentages of total motile sperm. (**B**) Percentages of progressively motile sperm. White bars (CONTROL): straws thawed following the conventional protocol (38 °C for 40 s in a water bath). Medium grey bars (PHOTO): straws thawed at 20 °C through irradiation with red light. Dark grey bars (ET): straws thawed at 20 °C inside the irradiation chamber but without any exposure to light. Results are expressed as box and whiskers; thus, median and quartiles (1st and 3rd) rather than means ± SEM are shown for each experimental point (Control, PHOTO, and ET) (n = 16). Different letters indicate significant (*p* < 0.05) differences between groups.

**Figure 4 animals-15-03353-f004:**
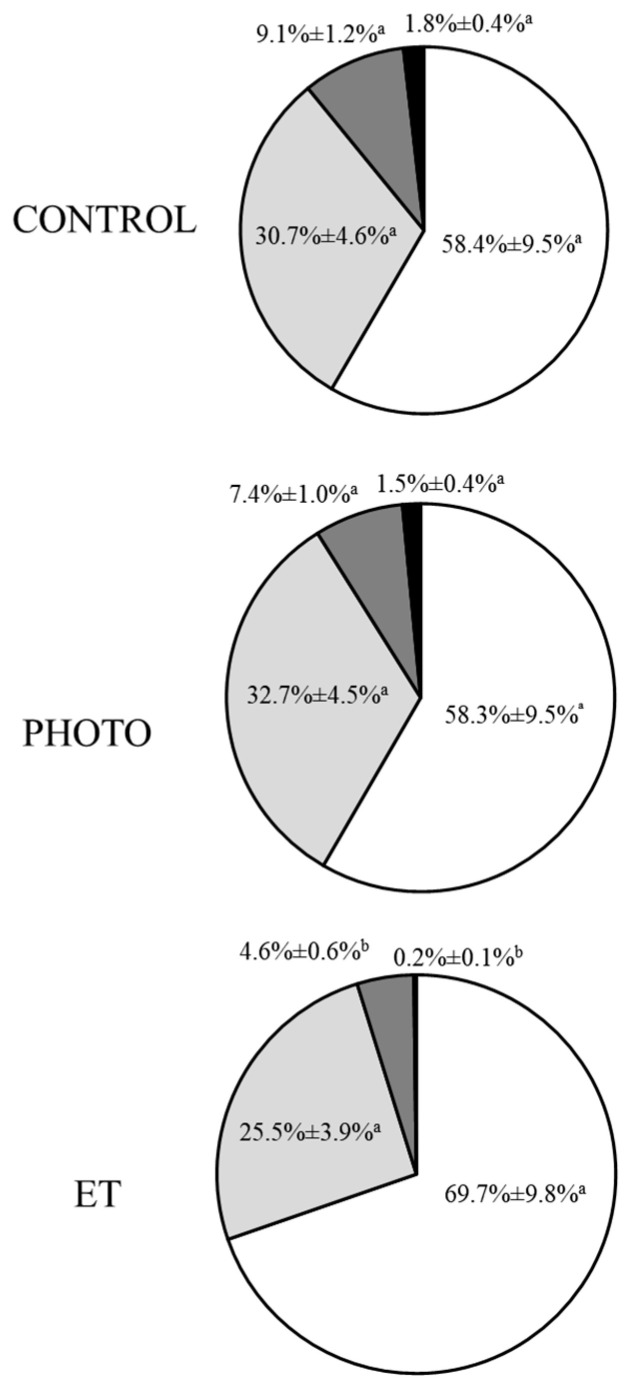
Motile sperm subpopulations in cryopreserved samples thawed through the conventional protocol, irradiation with red light at 20 °C, or thawed at 20 °C without irradiation. CONTROL: straws thawed following the conventional protocol (38 °C for 40 s in a water bath). PHOTO: straws thawed through irradiation with red light at room temperature (20 °C). ET: straws thawed at room temperature (20 °C), inside the irradiation chamber, but without any exposure to light. White sections: Subpopulation 1. Light grey sections: Subpopulation 2. Dark grey sections: Subpopulation 3. Black sections: Subpopulation 4. Results are expressed as mean ± SEM (n = 16). Different letters indicate significant (*p* < 0.05) differences between groups.

**Figure 5 animals-15-03353-f005:**
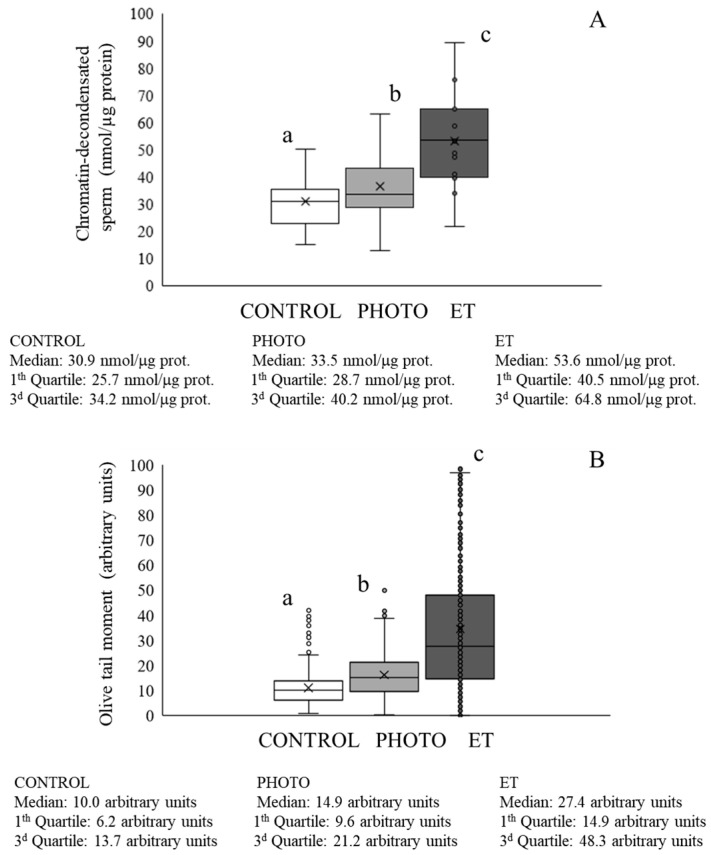
Chromatin decondensation and DNA fragmentation of cryopreserved sperm thawed through the conventional protocol, irradiation with red light at 20 °C, or thawed at 20 °C without irradiation. (**A**) Chromatin decondensation. (**B**) DNA fragmentation. White bars (CONTROL): straws thawed following the conventional protocol (38 °C for 40 s in a water bath). Medium grey bars (PHOTO): straws thawed at 20 °C through irradiation with red light. Dark grey bars (ET): straws thawed at 20 °C inside the irradiation chamber but without any exposure to light. Results are expressed as box and whiskers; thus, median and quartiles (1st and 3rd) rather than means ± SEM are shown for each experimental point (Control, PHOTO, and ET) (n = 16). Different letters indicate significant (*p* < 0.05) differences between groups.

**Figure 6 animals-15-03353-f006:**
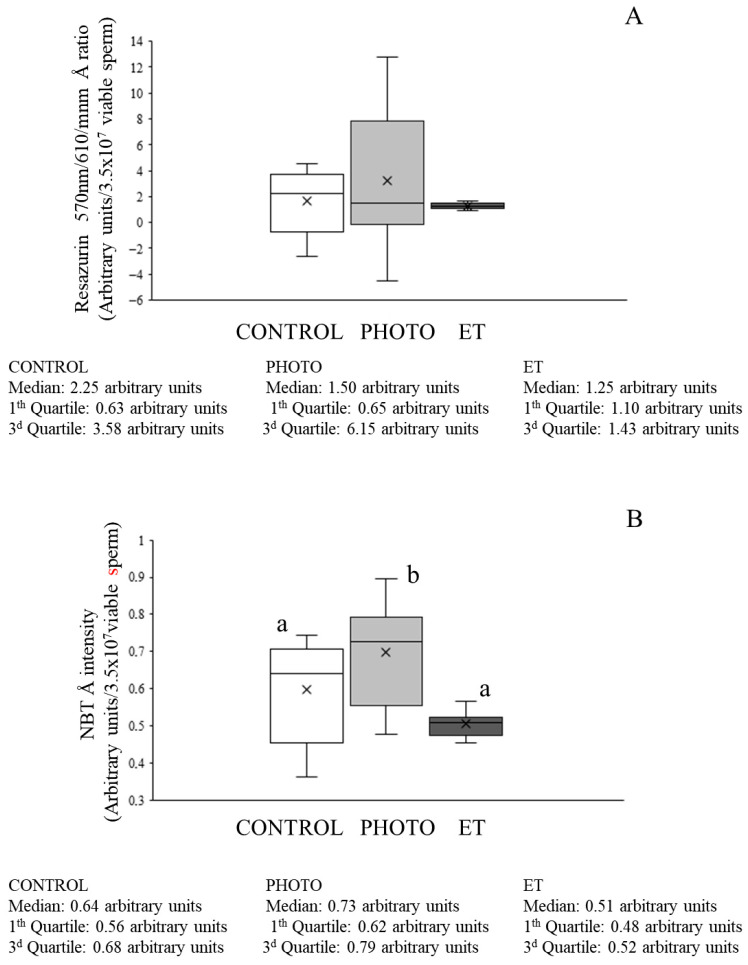
Resazurin ratios and absorbance intensities at 570 nm and 610 nm for NBT of cryopreserved sperm thawed through the conventional protocol, irradiation with red light at 20 °C, or thawed at 20 °C without irradiation. (**A**): resazurin ratios. (**B**): 570 nm/610 nm NBT intensities. White bars (CONTROL): straws thawed following the conventional protocol (38 °C for 40 s in a water bath). Medium grey bars (PHOTO): straws thawed at 20 °C through irradiation with red light. Dark grey bars (ET): straws thawed at 20 °C inside the irradiation chamber but without any exposure to light. Results are expressed as box and whiskers; thus, median and quartiles (1st and 3rd) rather than means ± SEM are shown for each experimental point (Control, PHOTO, and ET) (n = 16). Different letters indicate significant (*p* < 0.05) differences between groups.

**Figure 7 animals-15-03353-f007:**
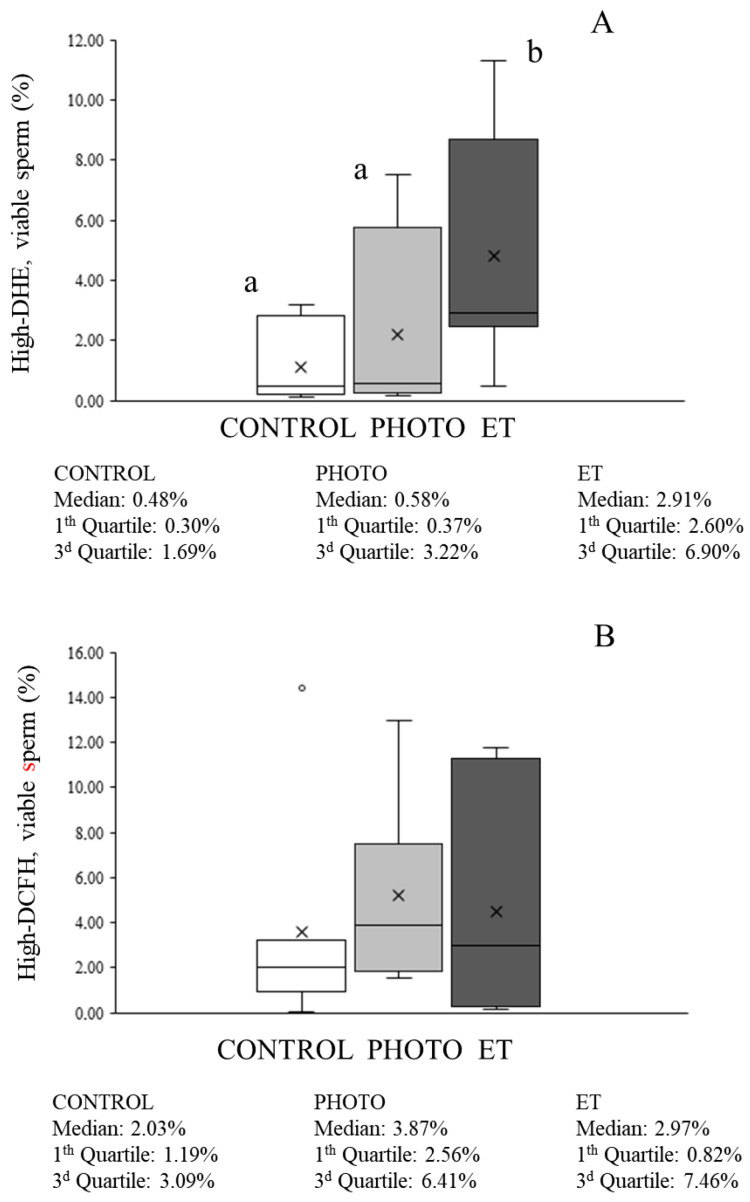
Percentages of viable sperm with high 2-E^+^OH and DCF intensities of cryopreserved sperm thawed through the conventional protocol, irradiation with red light at 20 °C, or thawed at 20 °C without irradiation. (**A**): 2-E^+^OH (superoxides). (**B**): DCF (total ROS). White bars (CONTROL): straws thawed following the conventional protocol (38 °C for 40 s in a water bath). Medium grey bars (PHOTO): straws thawed at 20 °C through irradiation with red light. Dark grey bars (ET): straws thawed at 20 °C inside the irradiation chamber but without any exposure to light. Results are expressed as box and whiskers; thus, median and quartiles (1st and 3rd) rather than means ± SEM are shown for each experimental point (Control, PHOTO, and ET) (n = 16). Different letters indicate significant (*p* < 0.05) differences between groups.

**Figure 8 animals-15-03353-f008:**
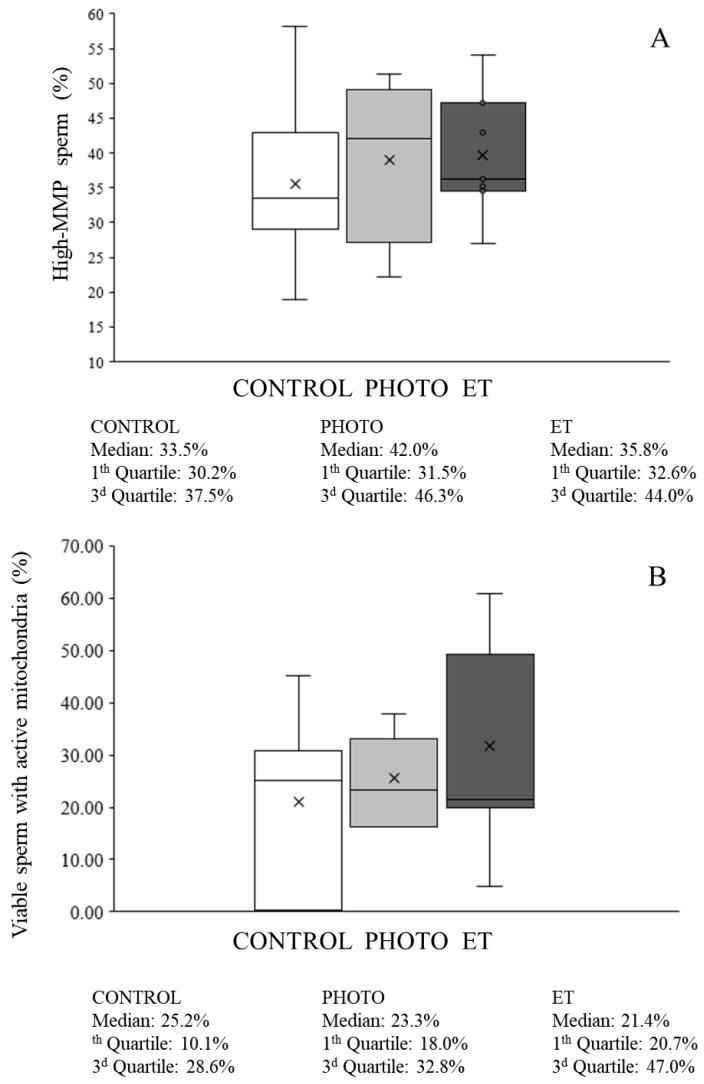
Percentages of sperm with high mitochondrial membrane potential and viable sperm with active mitochondria in cryopreserved sperm thawed through the conventional protocol, irradiation with red light at 20 °C, or thawed at 20 °C without irradiation. (**A**): sperm with high MMP. (**B**): viable sperm with active mitochondria following the MT-DR probe. White bars (CONTROL): straws thawed following the conventional protocol (38 °C for 40 s in a water bath). Medium grey bars (PHOTO): straws thawed at 20 °C through irradiation with red light. Dark grey bars (ET): straws thawed at 20 °C inside the irradiation chamber but without any exposure to light. Results are expressed as box and whiskers; thus, median and quartiles (1st and 3rd) rather than means ± SEM are shown for each experimental point (Control, PHOTO, and ET) (n = 16).

**Figure 9 animals-15-03353-f009:**
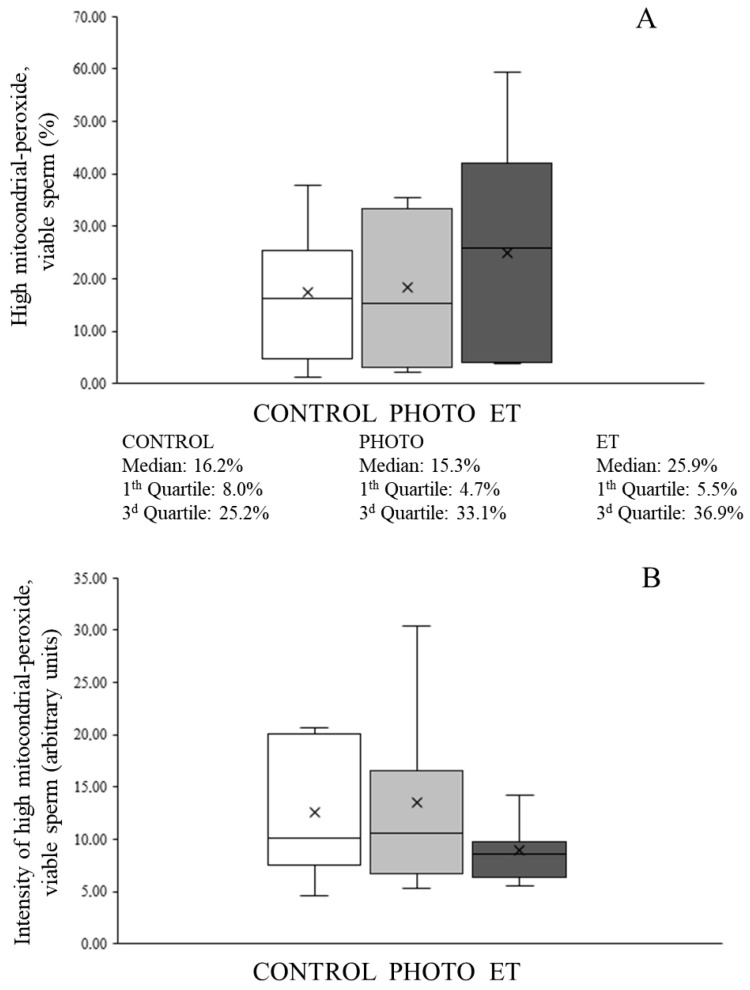
Sperm with high mitochondrial superoxide levels in cryopreserved sperm thawed through the conventional protocol, irradiation with red light at 20 °C, or thawed at 20 °C without irradiation. (**A**): Percentage of viable sperm with active mitochondria and high superoxide levels following the MitoSOX Red test. (**B**): Intensity of signal from viable sperm with active mitochondria and high superoxide levels following the MitoSOX Red test. White bars (CONTROL): straws thawed following the conventional protocol (38 °C for 40 s in a water bath). Medium grey bars (PHOTO): straws thawed at 20 °C through irradiation with red light. Dark grey bars (ET): straws thawed at 20 °C inside the irradiation chamber but without any exposure to light. Results are expressed as box and whiskers; thus, median and quartiles (1st and 3rd) rather than means ± SEM are shown for each experimental point (Control, PHOTO, and ET) (n = 16).

**Table 1 animals-15-03353-t001:** Mean motility parameters of cryopreserved sperm thawed through the conventional protocol (control), irradiation with red light at room temperature, or thawed at room temperature without irradiation (ET).

	CONTROL	PHOTO	ET
VCL (µm/s)	104.2 ± 0.6 ^a^	98.6 ± 0.7 ^c^	51.1 ± 1.9 ^b^
VSL (µm/s)	50.2 ± 0.7 ^a^	41.8 ± 0.8 ^c^	25.9 ± 1.0 ^b^
VAP (µm/s)	63.7 ± 0.6 ^a^	54.0 ± 0.7 ^c^	32.4 ± 1.1 ^b^
LIN (%)	50.6 ± 0.8 ^a^	45.4 ± 0.9 ^b^	50.9 ± 1.0 ^a^
STR (%)	77.1 ± 0.7	75.3 ± 0.8	79.8 ± 0.5
WOB (%)	63.2 ± 0.6 ^a^	57.8 ± 0.7 ^b^	63.4 ± 0.9 ^a^
mALH (µm)	4.16 ± 0.02 ^a^	4.45 ± 0.02 ^c^	2.38 ± 0.07 ^b^
BCF (Hz)	10.4 ± 0.2 ^a^	9.2 ± 0.2 ^b^	9.4 ± 0.2 ^ab^
DNC (µm^2^/s)	182.9 ± 22.0 ^a^	165.5 ± 10.9 ^a^	123.2 ± 8.3 ^b^
mDNC (%xµm)	143.6 ± 11.0 ^a^	141.2 ± 4.7 ^a^	121.0 ± 2.51 ^b^

Different superscripts in a row indicate significant (*p* < 0.05) differences. Results are expressed as mean ± SEM (n = 16).

**Table 2 animals-15-03353-t002:** Kinetic parameters of sperm populations observed in cryopreserved samples thawed through the conventional protocol (control), irradiation with red light at room temperature (irradiated), or thawed at room temperature without irradiation (ET).

	Subpopulation 1			Subpopulation 2			Subpopulation 3			Subpopulation 4		
	CONTROL	PHOTO	ET	CONTROL	PHOTO	ET	CONTROL	PHOTO	ET	CONTROL	PHOTO	ET
VCL (µm/s)	39.9 ± 0.3 ^a^	38.8 ± 0.4 ^a^	39.9 ± 0.9 ^ª^	86.4 ± 0.5 ^a^	82.0 ± 0.5 ^b^	73.4 ± 0.3 ^c^	125.6 ± 0.8 ^a^	116.5 ± 1.1 ^b^	114.2 ± 1.0 ^b^	161.1 ± 1.9 ^a^	157.3 ± 2.4 ^a^	217.2 ± 3.0 ^b^
VSL (µm/s)	23.3 ± 0.4 ^a^	21.5 ± 0.5 ^a^	20.3 ± 0.4 ^b^	53.2 ± 0.5 ^a^	44.9 ± 0.6 ^b^	39.8 ± 0.4 ^c^	66.0 ± 1.0 ^a^	51.4 ± 1.3 ^b^	48.2 ± 1.4 ^b^	54.3 ± 2.3 ^a^	49.2 ± 2.8 ^a^	24.5 ± 2.0 ^b^
VAP (µm/s)	28.2 ± 0.4 ^ª^	26.4 ± 0.4 ^ab^	25.9 ± 0.4 ^b^	60.0 ± 0.5 ^a^	51.7 ± 0.6 ^b^	47.6 ± 0.4 ^c^	77.7 ± 0.9 ^a^	63.1 ± 1.2 ^b^	63.5 ± 1.2 ^b^	82.9 ± 2.0 ^a^	74.9 ± 2.5 ^a^	49.5 ± 1.5 ^b^
LIN (%)	55.7 ± 0.5 ^a^	54.1 ± 0.5 ^a^	48.0 ± 0.6 ^b^	60.1 ± 0.6 ^a^	53.9 ± 0.7 ^b^	53.8 ± 0.7 ^b^	51.0 ± 1.2 ^a^	43.0 ± 1.5 ^b^	41.4 ± 1.4 ^b^	33.4 ± 2.6 ^a^	30.7 ± 3.3 ^a^	11.3 ± 1.2 ^b^
STR (%)	76.2 ± 0.4 ^a^	75.0 ± 0.5 ^a^	70.5 ± 0.4 ^b^	86.1 ± 0.6 ^a^	84.2 ± 0.6 ^a^	81.3 ± 0.5 ^b^	82.4 ± 1.0 ^a^	78.9 ± 1.3 ^a^	73.5 ± 1.12 ^b^	64.1 ± 2.4 ^a^	63.1 ± 3.0 ^a^	49.5 ± 1.4 ^b^
WOB (%)	70.1 ± 0.3 ^a^	68.5 ± 0.4 ^a^	63.2 ± 0.4 ^b^	68.3 ± 0.5 ^a^	62.4 ± 0.5 ^b^	64.6 ± 0.5 ^b^	60.6 ± 0.8 ^a^	53.2 ± 1.1 ^b^	55.2 ± 1.2 ^b^	50.8 ± 1.9 ^a^	46.9 ± 2.4 ^a^	22.8 ± 1.6 ^b^
mALH (µm)	1.86 o ± 0.01 ^a^	1.91 ± 0.02 ^a^	1.97 ± 0.03 ^a^	3.22 ± 0.02 ^a^	3.42 ± 0.02 ^b^	3.20 ± 0.02 ^ª^	4.83 ± 0.03 ^a^	5.19 ± 0.04 ^b^	5.01 ± 0.05 ^b^	6.97 ± 0.08 ^a^	7.27 ± 0.10 ^a^	13.85 ± 0.15 ^b^
BCF (Hz)	6.9 ± 0.1 ^a^	6.2 ± 0.1 ^b^	6.6 ± 0.2 ^ab^	11.3 ± 0.1 ^a^	10.6 ± 0.1 ^b^	9.9 ± 0.1 ^b^	12.2 ± 0.2 ^a^	10.3 ± 0.3 ^b^	10.1 ± 0.4 ^b^	10.3 ± 0.5 ^a^	9.7 ± 0.6 ^a^	1.0 ± 0.1 ^b^
DNC (µm^2^/s)	78.6 ± 1.5 ^a^	79.3 ± 1.7 ^a^	81.1 ± 2.0 ^a^	277.3 ± 2.0 ^a^	279.8 ± 2.3 ^a^	236.8 ± 2.0 ^b^	600.6 ± 3.7 ^a^	599.5 ± 4.8 ^ab^	576.3 ± 4.5 ^b^	1120.9 ± 8.4 ^a^	1140.1 ± 10.6 ^a^	3007.7 ± 22.1 ^b^
mDNC (%xµm)	97.8 ± 1.0 ^a^	94.4 ± 1.1 ^a^	88.7 ± 1.3 ^b^	183.7 ± 1.3 ^a^	175.3 ± 1.5 ^b^	164.1 ± 1.5 ^b^	235.0 ± 2.5 ^a^	214.3 ± 3.2 ^b^	201.8 ± 3.4 ^c^	220.9 ± 5.6 ^a^	215.5 ± 7.1 ^a^	156.1 ± 6.7 ^b^

Different superscripts in a row indicate significant (*p* < 0.05) differences between the three groups within the same subpopulation. Results are expressed as mean ± SEM (n = 16), corresponding to 14,278 motile sperm.

**Table 3 animals-15-03353-t003:** Environmental conditions and pregnancy rates (PR) of farms involved in this study. AI outcomes as PRs are provided for the control (CONTROL) and cryopreserved sperm thawed through irradiation with red LED light (PHOTO).

Trial	Geographical Coordinates of Farm	Altitude Above the Sea (m)	Temperature (°C)	Relative Humidity (%)	Animal Type	n Control	n Photo	PR Control (%)	PR Photo (%)	Variation in NRRs (%)
1 (Farm 1)	40°54′36″ S/73°36′58″ W	214	19	80	Angus cows	81	35	45.7 (37/81)	60.0 (21/35)	+31.3
2 (Farm 2)	39°46′42″ S/73°14′30″ O	15	17	85	Dairy Holstein cows	113	65	52.2 (59/113)	66.1 (43/65)	+26.6
Overall	-	-	-	-	-	194	100	49.4 (96/194)	64.0 (64/100)	+29.5

PR: pregnancy rates. CONTROL: animals inseminated with straws thawed under standard conditions at 38 °C for 45 s. PHOTO: animals inseminated with straws thawed through irradiation with red LED light. n: number of animals. Overall: global numbers considering the outcomes in all farms. In PR columns, data between brackets indicate the number of positive gestating cows against the total number of inseminated animals.

## Data Availability

Data are not publicly archived. However, all data are available upon request to the authors (J.E.R.-G., M.Y., A.R.-R., and D.B.).

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
