# Peer review of "Red LED Light Irradiation Increases the Resistance Against Environmental Stress of Frozen Bovine Sperm Thawed in Suboptimal Conditions"

_animals, 2025, doi:10.3390/ani15223353_

Round 1

Reviewer 1 Report

Comments and Suggestions for Authors

This study provides an innovative approach to improving thawing resilience of bovine sperm via red LED irradiation. While preliminary data are intriguing, the biological basis remains speculative and the fertility trial insufficient for firm conclusions. Refinement of the irradiation protocol, inclusion of biochemical markers (ATP, cytochrome oxidase, lipid peroxidation), and robust statistical validation are essential before commercial application. Major Revision Required. The work is promising but requires additional mechanistic and field-level evidence to substantiate the claimed benefits.

Title and Abstract

The title appropriately reflects the experimental focus and outcomes. However, the abstract presents excessive methodological detail (e.g., thawing times and light intervals). The claim of “putative effects on in vivo fertilizing ability” should be cautiously phrased given that the AI trial was preliminary with non-statistical data.
Introduction

The introduction establishes a clear rationale regarding sperm cryodamage and the relevance of thawing protocols. Nonetheless, the justification for using red LED light in bovine semen lacks sufficient mechanistic background.
Materials and Methods

Following concerns should be raised:

Pooling of 4–5 straws per bull before analysis may mask inter-bull variability.

The sample size (n=16 bulls) is acceptable, but the absence of replication across batches restricts statistical power.

The LED irradiation protocol (2’-1’-2’) appears arbitrary, with no explanation of wavelength intensity or fluence rate (mW/cm²).
Results

Results are detailed and data-rich, but interpretation often extends beyond statistical support. For instance:

Viability decreased in PHOTO samples, yet this reduction is not discussed with context to biology.

Increased ROS-positive viable sperm contradicts improved DNA integrity….. Need explaination.
Discussion

The discussion effectively relates findings to previous studies but occasionally drifts into overinterpretation. The claim that red light may “increase sperm resistance to environmental stress” lacks molecular substantiation, as mitochondrial potential remained unchanged. Moreover, citing photobiomodulation pathways without supporting biochemical data (e.g., ATP, cytochrome oxidase activity) weakens the argument.  The AI experiment is too limited to draw conclusions. Only two farms were used, and groups were unbalanced.

Author Response

REVIEWER 1

Title and Abstract

The title appropriately reflects the experimental focus and outcomes. However, the abstract presents excessive methodological detail (e.g., thawing times and light intervals). The claim of “putative effects on in vivo fertilizing ability” should be cautiously phrased given that the AI trial was preliminary with non-statistical data.

Thank you for your advice. Regarding your comment about the “putative effects on in vivo fertilizing ability”, this paragraph has been deleted from the title

Introduction

The introduction establishes a clear rationale regarding sperm cryodamage and the relevance of thawing protocols. Nonetheless, the justification for using red LED light in bovine semen lacks sufficient mechanistic background.

The study on mechanisms by which red LED light can affect sperm function is not fully displayed and, hence, there is a certain lack of information. Despite this, in the last years our working team has published several articles regarding this point, although there were focused in other species than bovine such as porcine, equine and asinine. Thus, whereas in porcine, equine and asinine freshly collected sperm the effect of the irradiation with red LED light seems to be related to the modulation of mitochondrial electron chain (Catalán J, Papas M, Gacem S, Mateo-Otero Y, Rodríguez-Gil, JE, Miró J, Yeste M. 2020. Biology 9: 254; Catalán J, Papas M, Trujillo-Rojas L, Blanco-Prieto O, Bonilla-Correal S, Rodríguez-Gil, JE, Miró J, Yeste M. 2020. Front. Cell Develop. Biol., 8: 8588261; Blanco-Prieto O, Maside C, Peña A, Ibáñez-Príncep J, Bonet, S, Yeste M, Rodríguez-Gil, J.E. 2022. Frontiers Cell Develop. Biol 10:930855), in frozen equine samples, red LED light improves resistance to thawing without any significant effect on MMP (Catalán J, Llavanera M, Bonilla-Correal S, Papas M, Gacem S, Rodríguez-Gil JE, Yeste M, Miró J. 2020. Theriogenology 157: 85-95). In our opinion, this discrepancy could be linked with the very different functional status that have freshly obtained semen samples that those subjected to freezing-thawing. Thus, one well known effect of freezing-thawing on sperm function is a prominent alteration of mitochondrial function linked to alterations of the electron chain, which is translated into a loss of mitochondrial activity (see Flores E, Fernández-Novell JM, Peña A, Rodríguez-Gil JE. 2009. Theriogenology, 72: 784-797 as an example in boar). The alteration in mitochondrial function, which is also linked to changes in the specific motile sperm subpopulation structure (Flores E, Fernández-Novell JM, Peña A, Rodríguez-Gil, J.E. 2009. Theriogenology, 72: 784-797; Catalán J, Papas M, Gacem S, Mateo-Otero Y, Rodríguez-Gil, JE, Miró J, Yeste, M. 2020. Biology 9: 254) would lead mitochondrial electron chain to a loss of sensitivity to light, since the freezing-thawing linked alterations would affect the light response.  Otherwise, there are other possible mechanisms by which red LED light can act on sperm function. Among them, the best known should be an opsin-related action, since opsins like rhodopsin have been found in mammalian sperm (Pérez-Cerezales S, Boryshpolets S, Afanzar O, Brandis A, Nevo R, Kiss V, Eisenbach M. 2015. Sci. Rep. 5, 16146). In boar sperm, the inhibition of the PKC-modulated opsin signaling pathway has a minor effect on the response to red LED light when compared with the involvement of mitochondrial electron chain (Blanco-Prieto O, Maside C, Peña A, Ibáñez-Príncep J, Bonet S, Yeste M, Rodríguez-Gil, J.E. 2022. Frontiers Cell Develop. Biol 10:930855). However, this doesn’t imply the absolute lack of an opsin-related light effect, and, in this way, opsin pathways could reach prominence when sperm mitochondrial activity was altered as in freezing-thawing conditions. There are other possible mechanisms that would be less affected by freezing-thawing that could increase its importance when mitochondrial activity was affected. Thus, direct action of red light on chemical bonds such as phosphodiester ones present in molecules like ATP has been described, increasing this effect the ability of phosphodiester bonds to accumulate energy (Deng J, Bezold D, Jessen H J, Walther A. 2020. Angew. Chem. Int. Ed. Engl. 59, 12084–12092). In this way, the sum of mechanisms that are minoritarian in fresh conditions would achieve paramount importance when mitochondrial activity was altered by phenomena like freezing-thawing; explaining thus the results show here, but also in previously published works (see Catalán J, Llavanera M, Bonilla-Correal S, Papas M, Gacem S, Rodríguez-Gil JE, Yeste M, Miró J. 2020. Theriogenology 157: 85-95). At this moment, this is only a hypothesis that needs much more, intensive work to be completely elucidated. In fact, this additional work clearly exceeds the main objective of this manuscript. In this way, a succinct approach to that has been added to both the Introduction and the Discussion to answer your kind advice.

Materials and Methods

Following concerns should be raised:

Pooling of 4–5 straws per bull before analysis may mask inter-bull variability.

Probably we have not explained ourselves well. In fact, we pooled 4-5 straws from the same ejaculate, not by mixing samples from diferent bulls, Text has been modified to clarify this mistake.

The sample size (n=16 bulls) is acceptable, but the absence of replication across batches restricts statistical power.

We are sorry but we don’t understand well this query. Perhaps is it related with the former question?  Our data treament was made taking into account a compromise to avoid masking of inter-bull variability avoiding the mixture of ejaculates from diferent bulls, and ev en from diferent ejaculates of the same bull. In this way, the same sample was utilized for determining all paràmetres in a separate manner, i.e., a pool of straws from the same ejaculate was utilized to analyze, viability,, motility, cytometric paràmetres, DNA fragmentation and condensation and so on. Sicerely, we think that this is the best manner in which statistical power was strength taking into account the unavoidable inter-samples variability.

The LED irradiation protocol (2’-1’-2’) appears arbitrary, with no explanation of wavelength intensity or fluence rate (mW/cm²).

In fact, the assumption of the utilized irradiation protocol was based upon our previous information regarding the most effective irradiation protocols applied to separate especies. After analyzing the information of all these studies, we opted to utilize a consensus protocol that yield optimal results in our previous works. Otherwise, wavelenght intensity was of 5.66 mW/cm2. Information regarding intensity has been added to the text.   

Results

Results are detailed and data-rich, but interpretation often extends beyond statistical support. For instance:

Viability decreased in PHOTO samples, yet this reduction is not discussed with context to biology.

Increased ROS-positive viable sperm contradicts improved DNA integrity….. Need explaination.

As you kindly indicated, this work is very rich in data. This richness often complicate interpretation, considering that we have not a complete picture of all molecular and biological interactions that play a role in sperm function. Thus, we have not an easy explanation for viability results in PHOTO samples, because light could affect cell membrane structure without launching apoptotic-like changes through many ways including photosentitive membrane receptors like opsins and other that we don’t know at this moment. Likewise, it is true that high ROS levels affect DNA integrity. However, the extension of damage will depend on many factors like the specific increase in ROS levels, which was not very high in our experimental model indeed, and the basal DNA fragmentation levels observed in frozen-thawed samples, which were higher than those of freshly obtained samples. Moreover, other factors like the specific activity of ROS scavenger mechanisms in thawed samples will be another key factor to explain

Discussion

The discussion effectively relates findings to previous studies but occasionally drifts into overinterpretation. The claim that red light may “increase sperm resistance to environmental stress” lacks molecular substantiation, as mitochondrial potential remained unchanged. Moreover, citing photobiomodulation pathways without supporting biochemical data (e.g., ATP, cytochrome oxidase activity) weakens the argument.  The AI experiment is too limited to draw conclusions. Only two farms were used, and groups were unbalanced.

Of course, you are right, especially concerning IA results. In this sense, we tried to refrain from conclusions. However, following your advice, it seems to be clear that this effort has not been enough. In this way, we have further modified the text to make clearer all caveats and trying to avoid overinterpretation of data.

Reviewer 2 Report

Comments and Suggestions for Authors

The manuscript entitled “Red LED light irradiation increases the resistance against environmental stress of frozen bovine sperm thawed in suboptimal conditions: putative effects on in vivo fertilizing ability”. This study aimed 1): to investigate whether thawing in suboptimal conditions commercial, cryopreserved bovine sperm through irradiation with red LED light could improve the resilience of cells to the induced thawing-related environmental stress. 2) exploring the possibility that red LED irradiation could have an impact not only on post-thawing sperm quality and functionality, but also on in vivo reproductive performance after AI in farm conditions. PHOTO samples significantly improved sperm quality parameters. Besides, PHOTO straws yield greater pregnancy rates (64.0% vs. 49.4% in CONTROL) when evaluated in 2 different farms.

General comments:

The manuscript is well written however it lacks some details.

The present manuscript rises many issues:

  • The experimental design is not well described. The authors made a lot of efforts to describe methods. For example, we do not know what the number of total straws was provided by the IA center. If the number of bulls was 16, using three treatments 5 to 6 straws and three treatments we will need 240 straws.
  • The number of inseminated females is not provided in MM section. The reared has to go the results to understand that the total was 294.
  • I do not understand why the authors introduce another factor the parity of females?? Two farms are additional factors, I am sure, that the rearing management in each farm.
  • They authors have to specify that after the first experiment, they discarded the treatment ET from their AI trials.

Line 43: . Viability of PHOTO samples was significantly decreased PLZ replace by Viability of PHOTO samples decreased significantly

Line 49: , more in vivo studies should be needed to establish this statement PLZ , more in vivo studies are needed to establish this statement.

protocol set in [13] the authors should add the reference

according to [14,15] PLZ add the references (Name of authors)

The authors are invited to fellow the same logique throughout the manuscript, they presented Control, PHOTO and ET, and sometine they started by PHOTO

The authors should writ their treatments in the same way: they wrote ET and RT, they wrote PHOTO and Photo

Author Response

REVIEW 2

Comments and Suggestions for Authors

The manuscript entitled “Red LED light irradiation increases the resistance against environmental stress of frozen bovine sperm thawed in suboptimal conditions: putative effects on in vivo fertilizing ability”. This study aimed 1): to investigate whether thawing in suboptimal conditions commercial, cryopreserved bovine sperm through irradiation with red LED light could improve the resilience of cells to the induced thawing-related environmental stress. 2) exploring the possibility that red LED irradiation could have an impact not only on post-thawing sperm quality and functionality, but also on in vivo reproductive performance after AI in farm conditions. PHOTO samples significantly improved sperm quality parameters. Besides, PHOTO straws yield greater pregnancy rates (64.0% vs. 49.4% in CONTROL) when evaluated in 2 different farms.

General comments:

The manuscript is well written however it lacks some details.

The present manuscript rises many issues:

  • The experimental design is not well described. The authors made a lot of efforts to describe methods. For example, we do not know what the number of total straws was provided by the IA center. If the number of bulls was 16, using three treatments 5 to 6 straws and three treatments we will need 240 straws.

Dear reviewer. Your assertion is exact. Fortunately, we worked with a big AI center (the CIA of the Austral University of Chile) that has at disposal of investigator a great number of samples becoming from a great number of bulls that have been in the Center in the past two decades. Thus, there are individual ejaculates with a great number of straws that were at our disposal for use in the experiments. In this way, the exact number of straws that were utilized (ignoring straws thate were not finally utilized) was 260. This has been added to the text.

  • The number of inseminated females is not provided in MM section. The reared has to go the results to understand that the total was 294.

We apologize for the lack of information. This value has been added to the text.

  • I do not understand why the authors introduce another factor the parity of females?? Two farms are additional factors, I am sure, that the rearing management in each farm.

You are right when indicated that the rearing management of each farm has a key effect on the specific reproductive performance, and that management can have more important than parity. Notwithstanding, we have added parity, either heifers or multiparous under an informative basis, since parity can also have an influence on the reproductive capacity of a cow, albeit less important than management. However, it is obvious that this information seems to induce some confusion, even more so considering the inherent very preliminary and, hence, without no statistical value, of results. Thus, we have opted to delete the information regarding parity from tables, mixing thus results from heifers and multiparous in each farm.

  • They authors have to specify that after the first experiment, they discarded the treatment ET from their AI trials.

This information has been added to the text.

Line 43: . Viability of PHOTO samples was significantly decreased PLZ replace by Viability of PHOTO samples decreased significantly

Done

Line 49: , more in vivo studies should be needed to establish this statement PLZ , more in vivo studies are needed to establish this statement.

Done

protocol set in [13] the authors should add the reference

Name of authors has been added to the text.

according to [14,15] PLZ add the references (Name of authors)

Name of autors added.

The authors are invited to fellow the same logique throughout the manuscript, they presented Control, PHOTO and ET, and sometine they started by PHOTO

We have modified the text following your advice, excepte in a few cases in which this order would alter the logical of the paragraph.

The authors should writ their treatments in the same way: they wrote ET and RT, they wrote PHOTO and Photo

We have revised the text according to your advice. We expect that no mistakes were left regarding this point now.

Reviewer 3 Report

Comments and Suggestions for Authors

I thank the authors for their research. I have the following comments on the manuscript:
1. In this version, the text line numbers after the graphic abstract are missing, which makes it difficult to comment.
2. The graphic abstract needs minor adjustments to the position of the ">" symbols.
3. In the abstract, and especially in the "Introduction" section, the authors highlight the problem of "suboptimal conditions for thawing insemination doses," meaning thawing at room temperature (approximately 20 degrees Celsius). However, the text of this manuscript contains no clear confirmation that such a suboptimal thawing method is actually used on farms (regionally?). For example, on farms in my country, when artificially inseminating cows with commercial doses, the standard (optimal) thawing method, using a water bath at 38-40 degrees Celsius, has been used since long time ago. I don't see this method as any difficult to use (given the widespread availability and affordability of water baths or other similar thermos-defrosting devices). I believe that if some regions still thaw using a suboptimal methodology, the primary focus should be on widespread adoption and/or standardization of the optimal thawing method, rather than on ways to mitigate the negative impact of inappropriate thawing methods. In this regard, I'm wondering whether red light irradiation can improve the results of the optimal thawing method.
4. The authors mention a "concentration" of 10-20 million sperm cells in their insemination doses. Does this number of sperm cells refer to a single insemination dose, i.e., per straw? 
5. It would be helpful to provide technical specifications (or a photograph) of the device for irradiating the straws with red light.
6. It is unclear from the description how the temperature change in the straws during thawing was measured. Please provide a detailed explanation in the text. Furthermore, as can be seen in Figure 1, thawing the straws with red light irradiation resulted in a change in the thawing curve on the graph (compared to thawing under the same conditions but without red light). Is it possible to simulate this curve without using red light to determine the effect of thawing rate itself on the result?
7. The text of the manuscript frequently uses the name "RT" for one of the experimental groups instead of "ET." Please select one name and make corrections so that the group names in the text are always consistent.
8. Section 2.2: Delete the introductory sentence.
9. Section 2.2.3.: Which symbol denoted by 3 does this mean: "VAP ³ 10 μm/s" and "STR ³ 70%"? Please correct.
10. Sections 2.2.6 and 2.3.1.: Check the correct units of measurement (mL or microL?)
11. Section 2.4.: It is unclear whether same batch (from the same bull, frozen on the same day under the same conditions) commercial doses were used to inseminate the two groups of animals (control versus test)? I assume that the same batch doses were used. Please provide a clearer description in the text.
12. Figure 1: 37 or 38 degrees? The information in the figure should match the information in the legend/text of the manuscript.
13. Table 2: It is difficult to read in its current form.
14. Section 3.5: The term "olive tail moment" is mentioned. This term is not found anywhere else in the text. Please provide an explanation.
15. Figure 6B: Correct the information on the Y-axis.
16. The "Discussion" section does not provide a clear explanation for the observed interesting phenomenon (namely, that doses thawed with red light irradiation, compared with doses thawed optimally, yielded better in vivo results, despite the fact that the in vitro quality parameters of these doses were worse than those of doses thawed optimally). 
17. Suppl. Fig. 1: This supplement is not mentioned anywhere in the manuscript.

Author Response

REVIEW 3

Comments and Suggestions for Authors

I thank the authors for their research. I have the following comments on the manuscript:
1. In this version, the text line numbers after the graphic abstract are missing, which makes it difficult to comment.

We apologize for the mistake. Line numbers have been added in the whole text.

  1. The graphic abstract needs minor adjustments to the position of the ">" symbols.

“>” positions have been relocated. This trouble and the another above mentioned were caused during the transformation of the text to PDF format. We expect that this had been solved now.

  1. In the abstract, and especially in the "Introduction" section, the authors highlight the problem of "suboptimal conditions for thawing insemination doses," meaning thawing at room temperature (approximately 20 degrees Celsius). However, the text of this manuscript contains no clear confirmation that such a suboptimal thawing method is actually used on farms (regionally?). For example, on farms in my country, when artificially inseminating cows with commercial doses, the standard (optimal) thawing method, using a water bath at 38-40 degrees Celsius, has been used since long time ago. I don't see this method as any difficult to use (given the widespread availability and affordability of water baths or other similar thermos-defrosting devices). I believe that if some regions still thaw using a suboptimal methodology, the primary focus should be on widespread adoption and/or standardization of the optimal thawing method, rather than on ways to mitigate the negative impact of inappropriate thawing methods. In this regard, I'm wondering whether red light irradiation can improve the results of the optimal thawing method.

As you kindly indicated, farming conditions are not the same worldwide. For instance, environmental temperatures in countries such as Chile, Spain or Italy can vary enormously, with areas in which temperature can range from -8ºC in winter to 45ºC in summer. Moreover, although devices such as thermos-defrosting ones are available, they are not universally utilized by practitioners, who in too many cases are working in the field in the absence of temperature-controlled water baths and similar. It is worth noting that, although regular thermos can maintain the temperature for a while, most of them don’t have dynamic systems to maintain a fixed temperature indefinitely, whereas the transport and further use of water baths is not practical in many farms and situations. In this way, although a tight control of temperature is highly desirable and can be applied in practice, the reality is that in many world areas this control is not feasible to all practitioners. Thus, the appearance cheap, practical systems that could aid in reinforcing a stricter control of thawing temperatures could be desirable for increasing reproductive performance in not so-optimal farming conditions.

  1. The authors mention a "concentration" of 10-20 million sperm cells in their insemination doses. Does this number of sperm cells refer to a single insemination dose, i.e., per straw? 

Yes, this is the sperm concentration per straw.

  1. It would be helpful to provide technical specifications (or a photograph) of the device for irradiating the straws with red light.

The device utilized in this study is a prototype system that has not been commercialized. This implies that its appearance would be very different to that of a commercial device more adapted for in-farm work. Thus, we have opted to include in the text the most important technical specifications, namely, the wavelenght intensity, which was 5.66 mW/cm2. Otherwise, if you would want to receive more information regarding how straws are placed inside the system and the general aspect of prototype, don’t hesitate to indicate it to us and we will send you it without restrictions.

  1. It is unclear from the description how the temperature change in the straws during thawing was measured. Please provide a detailed explanation in the text. Furthermore, as can be seen in Figure 1, thawing the straws with red light irradiation resulted in a change in the thawing curve on the graph (compared to thawing under the same conditions but without red light). Is it possible to simulate this curve without using red light to determine the effect of thawing rate itself on the result?

A more detailed description of temperature curves has been added. You are right that switching on the irradiation system induced a noticeable increase in the initial thawing velocity, which reaches 0ºC at about 60 s of thawing, whereas in ET samples, the ramp reaches 0ºC at about 120 s. Subsequently, curves of both PHOTO and ET until reaching 20ºC are very similar. Unfortunately, we have not found any practical manner to simulate the curve following your advice. Thus, we are not able to determine the effect that you are kindly pointing out.

  1. The text of the manuscript frequently uses the name "RT" for one of the experimental groups instead of "ET." Please select one name and make corrections so that the group names in the text are always consistent.

Thank you very much for your advice. We have revised the ext according to your advice. We expect that no mistakes will be left regarding this point now.

  1. Section 2.2: Delete the introductory sentence.

Done

  1. Section 2.2.3.: Which symbol denoted by 3 does this mean: "VAP ³ 10 μm/s" and "STR ³ 70%"? Please correct.

This has been a mistake, probably linked to the transformation of the initial Word file to a PDF one. The “3” should be, actually, ”≤”. This has been corrected and we will vigilant to ensure that this change will not appear again.

  1. Sections 2.2.6 and 2.3.1.: Check the correct units of measurement (mL or microL?)

Checked and mistakes corrected.

  1. Section 2.4.: It is unclear whether same batch (from the same bull, frozen on the same day under the same conditions) commercial doses were used to inseminate the two groups of animals (control versus test)? I assume that the same batch doses were used. Please provide a clearer description in the text.

As you kindly indicate, same batch of doses from the same ejaculate were utilized for replicates in the AI trials. This information has been clarified into the text.

  1. Figure 1: 37 or 38 degrees? The information in the figure should match the information in the legend/text of the manuscript.

38ºC. This has been indicated in the text.

  1. Table 2: It is difficult to read in its current form.

Frankly, we ever have difficulties to display in an effective manner such a large amount of information in a simpler manner. We have tried other options, like splitting the Table into four sub-tables, one per each subpopulation. However, this manner multiplies the number of tables without yielding a better clarification of results. Thus, we opted to maintain the format of the Table, although if you have a better idea, please, we will be very glad to try it.

  1. Section 3.5: The term "olive tail moment" is mentioned. This term is not found anywhere else in the text. Please provide an explanation.

The “olive tail” refers to the area observed in each sperm after the determination of DNA fragmentation. As greater is the area, which forms a tail that vaguely remembers an olive form, greater is the DNA fragmentation level. A succinct description of the name has been added in the M&M section.

  1. Figure 6B: Correct the information on the Y-axis.

Done.

  1. The "Discussion" section does not provide a clear explanation for the observed interesting phenomenon (namely, that doses thawed with red light irradiation, compared with doses thawed optimally, yielded better in vivo results, despite the fact that the in vitro quality parameters of these doses were worse than those of doses thawed optimally). 

You are right considering our results. However, we didn’t include any reference regarding this point in Discussion since in vivo results are very preliminary and not conclusive, only indicative. Therefore, an in-depth discussion on this point would be inherently speculative. Otherwise, the easiest answer to this paradox would be perhaps that, after analyzing the absolute values of the analyzed in vitro parameters, the real number of sperm that are affected by freezing-thawing following results in vitro tests were not very high. In this way, whereas the absolute number of sperm that will colonize oviduct in in vivo conditions is low (about 17,000; see Hawk HW. J Dairy Sci 1987, 70(7),1487-1503. doi: 10.3168/jds.S0022-0302(87)80173-X), both CONTROL and PHOTO samples, have many thousands-to-millions of sperm with acceptable function characteristics, regardless of relative differences in the in vitro quality tests. In these conditions, if all other factors modulating the success of AI are optimally conducted (i.e., properly conducted fixed-time AI protocols, optimal AI application, etc.), the impact of differences observed in in vitro analysis should be not very intense. Otherwise, this can lead to another question, namely why in vivo results suggested an improving effect of photostimulation if considering that the actual number of acceptable function sperm colonizing oviduct was similar when comparing CONTROL and PHOTO. In this case, the explanation would be linked not to the absolute number of sperm arriving to oviduct, but to the function status of these cells. Thus, in species such as boar, photostimulation was able to increase the number of sperm that achieved the full capacitated status and, hence, increased the absolute number of sperm able to fertilize (Yeste M, Codony F, Estrada E, Lleonart M, Balasch S, Peña A, Bonet S, Rodríguez-Gil JE. Sci Rep 2016, 6, 22569. doi 10.1038/srep22569). In fact, this effect was related with concomitant increases of in vivo fertility and prolificacy, although these results were variable, depending on factors such as specific farm management and environmental conditions (Blanco Prieto O, Catalán J, Lleonart M, Bonet S, Yeste M & Rodríguez-Gil JE. Reprod Domest Anim 52019, 4(8):1145-1148. doi: 10.1111/rda.13470; Crespo S, Martínez M & Gadea J. Animals (Basel) 2021,11(6), 1656. doi: 10.3390/ani11061656). In this way and considering the possibility that a similar increase in sperm ability to reach capacitation is induced in bull sperm, although the total number of sperm that colonize oviducts was similar when compared CONTROL and PHOTO, the greater ability to reach capacitation   status would lead to a greater fertilizing ability efficiency after photostimulation. Following your advice, and admitting the need for more in-depth studies, this rationale has been included in the Discussion section. Articles cited here are included in the manuscript in case they were not already cited.

  1. Suppl. Fig. 1: This supplement is not mentioned anywhere in the manuscript.

This figure shows some representative plots regarding results of flux cytometer. A mention has been included in the text.        

Round 2

Reviewer 1 Report

Comments and Suggestions for Authors

The authors have adequately addressed all major and minor comments raised by me. The revisions have significantly improved the clarity, scientific rigor, and presentation of the manuscript. Based on the thorough revisions and the overall quality of the manuscript, I recommend this paper for acceptance.